# Structural basis for specific RNA recognition by the alternative splicing factor RBM5

Komal Soni [1,2], Pravin Kumar Ankush Jagtap [1,2], Santiago Martínez-Lumbreras[1,2], Sophie Bonnal [3], Arie Geerlof[1], Ralf Stehle[1,2], Bernd Simon[4], Juan Valcárcel[3,5] & Michael Sattler [1,2] ✉

The RNA-binding motif protein RBM5 belongs to a family of multi-domain RNA binding proteins that regulate alternative splicing of genes important for apoptosis and cell proliferation and have been implicated in cancer. RBM5 harbors structural modules for RNA recognition, such as RRM domains and a Zn finger, and protein-protein interactions such as an OCRE domain. Here, we characterize binding of the RBM5 RRM1-ZnF1-RRM2 domains to *cis*-regulatory RNA elements. A structure of the RRM1-ZnF1 region in complex with RNA shows how the tandem domains cooperate to sandwich target RNA and specifically recognize a GG dinucleotide in a non-canonical fashion. While the RRM1-ZnF1 domains act as a single structural module, RRM2 is connected by a flexible linker and tumbles independently. However, all three domains participate in RNA binding and adopt a closed architecture upon RNA binding. Our data highlight how cooperativity and conformational modularity of multiple RNA binding domains enable the recognition of distinct RNA motifs, thereby contributing to the regulation of alternative splicing. Remarkably, we observe surprising differences in coupling of the RNA binding domains between the closely related homologs RBM5 and RBM10.

Alternative splicing (AS) greatly expands the protein repertoire encoded by the genome, whereby alternatively spliced isoforms can be translated into proteins with distinct, often antagonistic functions. Dysregulation of AS events has been correlated with numerous human diseases, including cancer[1–5]. The regulation of AS is complex and involves recognition of *cis*-acting RNA elements in the pre-mRNA by trans-acting protein splicing factors, which can favor or impair the recruitment of core splicing factors to the correct splice sites[6,7].

RNA-binding motif protein 5 (RBM5, also known as H37 or LUCA-15) is a *trans*-acting RNA binding protein, which is down-regulated in a variety of cancers[8–11] and suggested to regulate metastasis by directly altering expression and activation of proteins important for this process[12,13]. RBM5 belongs to a larger protein family that includes the

related multidomain RNA binding proteins RBM6 and RBM10, which share a similar domain organization and exhibit high sequence similarity with RBM5 (30% and 50%, respectively) (Fig. 1a, b). Interestingly, the chromosomal region (3p21.3) comprising the *RBM5* and *RBM6*[14] genes is frequently deleted in genomes of heavy smokers and lung cancer patients, consistent with a role as tumor suppressors[15,16]. On the other hand, RBM5 was found to be up-regulated in ovarian and breast cancers[17,18], suggesting an important albeit complex mode of regulation by RBM5, likely indicating differential cell type-specific activities.

RBM5 is involved in regulation of AS of pancreatic adenocarcinoma-associated events[19] and of apoptosis-related genes including the cell-death receptor *Fas*[20] and the initiator *Caspase-2*[21]. In the latter, it promotes formation of the pro-apoptotic Caspase-2

[1]Helmholtz Munich, Molecular Targets and Therapeutics Center, Institute of Structural Biology, Ingolstädter Landstrasse 1, 85764 Neuherberg, Germany. [2]Technical University of Munich, TUM School of Natural Sciences, Department of Bioscience, Bavarian NMR Center, Lichtenbergstrasse 4, 85748 Garching, Germany. [3]Centre de Regulació Genòmica, Barcelona Institute of Science and Technology and Universitat Pompeu Fabra, Barcelona, Spain. [4]Structural and Computational Biology Unit, European Molecular Biology Laboratory, 69117 Heidelberg, Germany. [5]Institució Catalana de Recerca i Estudis Avançats, Barcelona, Spain. ✉e-mail: michael.sattler@helmholtz-munich.de

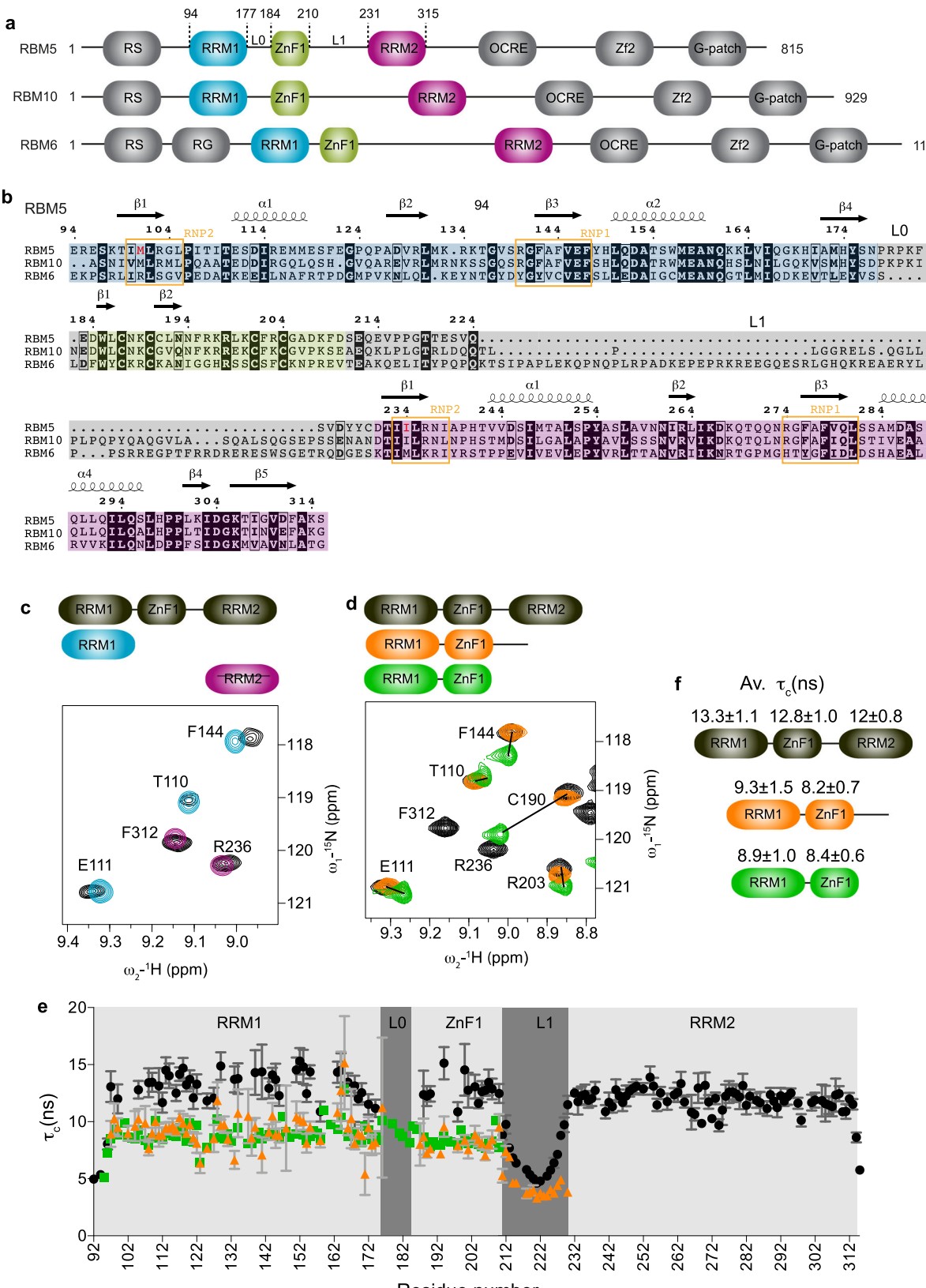

isoform by interacting with a U/C-rich element immediately upstream of the splicing repressor element ln100[21,22]. RBM5, RBM6, and RBM10 (Fig. 1a, b) are involved in AS regulation of *NUMB* pre-mRNA, where RBM5/6 and RBM10 have antagonistic effects[23]. *NUMB* encodes an inhibitor of NOTCH pathway, which is hyper-activated in ≈40% of human lung cancers and linked to breast cancer[24], making inhibition of

the NOTCH pathway a promising approach for cancer therapy[25–27]. Recently it was shown that RBM5 and RBM10 cross-regulate each other[28,29]. Altogether, these data indicate that despite the high similarities between RBM5, 6, and 10, the proteins can have distinct functional activities, and the underlying molecular and structural mechanisms are still poorly understood.

**Fig. 1 | Structural characterization of RBM5 RRM1-ZnF1 tandem domains.**
**a** Domain organization and **b** multiple sequence alignment of human RBM5, RBM6, and RBM10 proteins. RNP2 and RNP1 motifs of RRM1 and RRM2 are boxed in orange and domain boundaries of RRM1, ZnF1, and RRM2 are highlighted in blue, green, and pink, respectively. Linkers L0 and L1 connecting RRM1-ZnF1 and ZnF1-RRM2, respectively are highlighted in gray. The non-canonical hydrophobic residues found instead of the canonical aromatic residues in RNP2 motif of both RBM5 RRM1 and RRM2 domains are colored in red. **c** Overlay of $^1$H-$^{15}$N HSQC spectra of three RRM1-ZnF1$_S$-RRM2 domains construct (black) and single RRM1 (light blue) and

RRM2 (purple) domains. **d** Overlay of $^1$H-$^{15}$N-HSQC spectra of the three RRM1-ZnF1$_S$-RRM2 domains construct (black) with that of RRM1-ZnF1$_S$-L1 (orange) and RRM1-ZnF1$_S$ (green). Zoomed representative residues are shown. **e** Tumbling correlation time $\tau_c$ values (calculated from the ratio of $^{15}$N $R_2/R_1$ relaxation rates) are plotted vs. residue number for RRM1-ZnF1$_S$-RRM2 (black), RRM1-ZnF1$_S$-L1 (orange) and RRM1-ZnF1$_S$ (green) and their domain-wise average $\tau_c$ values ± SD are indicated in **f**. Error bars are derived from relaxation data. Source data are provided as a Source Data file.

RBM5 comprises multiple domains, including an arginine/serine rich (RS) region, two RNA recognition motif (RRM) domains (RRM1 and RRM2)[30], two zinc finger domains (ZnF1 and ZnF2)[31], an OCRE domain (OCtamer REpeat)[32] and a glycine-rich domain (G-patch) at the C-terminus (Fig. 1a). The RRM domains harbor two consensus motifs involved in RNA interaction known as RNP1 ([RK]-G-[FY]- [GA]- [FY]-[ILV]-X-[FY]) and RNP2 ([ILV]- [FY]- [ILV]-X-N-L)[30]. Notably, both of the RBM5 RRM domains are non-canonical as they lack the consensus aromatic residue in RNP2 (Fig. 1b), which might impact their RNA recognition.

Given the presence of multiple domains in the RBM5 family proteins it is unknown what the contributions of the different domains for RNA binding, interactions with other proteins and biological function of the protein are. Consistently, it has recently been shown that protein-protein interactions mediated via OCRE domain[33–35] are necessary in case of *Fas* AS regulation[20,35] while protein–RNA interactions via its RNA binding domains (RRM1 and RRM2) are important for *Caspase-2* AS regulation[36].

The N-terminal region of RBM5 contains three RNA binding domains, namely RRM1, ZnF1, and RRM2, connected together by linker regions (L0, 7 residues and L1, 20 residues, Fig. 1b), suggesting that distinct individual RNA binding preferences can contribute to the overall RNA binding of RBM5. In accordance, recent studies show that the RanBP2-type ZnF1 domain selectively binds to AGGGAA sequence motif with high nanomolar affinity[37] while RRM2 can recognize both CU and GA rich RNA sequences with low micromolar affinity[38]. CLIP-Seq[23] derived consensus motifs reflect previously identified U/C-rich RBM5 regulatory elements[21] and known ZnF1[37] and RRM2[38] binding motifs. In contrast, RNAcompete experiments identified consensus motifs that appear to mainly reflect the binding preferences of ZnF1 (GAAGGAA, GAAGGAG)[39]. Of note, the different top scoring motifs identified using CLIP-Seq (UCAUCGA, AGUAACG, AAGGAAAG, CAA-GAGUU, AUCUUUGU, CCGGGACA, among others)[23] show little sequence overlap, although the RNAcompete consensus motif is quite similar to one of the CLIP-Seq motifs (AAGGAAAG). There are some interesting similarities for the RNA binding preferences of RBM5 and its homolog RBM10. Similar to RBM5, the tandem RRM1-ZnF1 domains of RBM10 have been shown to recognize a CUGUGGA-motif[40], while CLIP-Seq[23] and PAR-CLIP[40,41] identified C-rich sequences shown to be preferentially bound by RRM2[40]. Solution NMR structures of the unbound RBM10 RRM1, ZnF1, and RRM2 domains (PDB accession codes: 2LXI, 2MXV, and 2M2B, respectively), and of the RBM5 ZnF1[42] and RRM2[38] domains have been reported. Given the variety of cognate RNA sequences reported it is important to dissect the structural mechanisms of how multiple RNA binding domains recognize these RNA ligands.

Here, we combined X-ray crystallography, solution NMR, small angle X-ray scattering (SAXS), and isothermal titration calorimetry (ITC) to study the structure, dynamics, and contribution of the three RNA binding domains of RBM5 to RNA recognition. A crystal structure of the tandem RRM1-ZnF1 domains in complex with RNA, reveals how the RNA is sandwiched between the two domains and recognized in a non-canonical way. SAXS and NMR analyses of the RRM1-ZnF1-RRM2 region show that the RRM1-ZnF1 domains act as a single module, while RRM2 is more dynamic. ITC binding data highlight the affinity

contribution of the individual domains which work together cooperatively. Using paramagnetic relaxation enhancement (PRE) NMR experiments coupled with SAXS, we unveil the dynamic interplay of the different RNA binding domains of RBM5 for RNA binding. Our findings highlight the distinct domain contributions towards RNA recognition of RBM5 and underline differences with its related homolog RBM10.

## Results

### Domain architecture of the RBM5 RNA recognition modules

To study how RBM5 recognizes *cis*-regulatory RNA motifs, we investigated the roles of the three RNA binding domains (RRM1-ZnF1-RRM2), which are connected by linkers L0 (7 residues, between RRM1 and ZnF1) and L1 (20 residues, between ZnF1 and RRM2) (Fig. 1a, b). To study the interplay between domains, we also analyzed the tandem RRM1-ZnF1 and the individual RRM1 and RRM2 domains.

First, we assessed whether inter-domain interactions exist between the different RNA binding domains of RBM5. We therefore expressed and purified $^{15}$N-labelled single RRM1 (residues 94-184) and RRM2 (residues 231−315) domains, as well as the tandem RRM1-ZnF1 (residues 94−210) and RRM1-ZnF1-RRM2 (residues 94−315) domains, both harboring a protein stabilizing point mutation C191G[43] (henceforth denoted as RRM1-ZnF1$_S$ and RRM1-ZnF1$_S$-RRM2). A superposition of $^1$H-$^{15}$N HSQC spectra of RRM1-ZnF1$_S$-RRM2 with those of the individual RRM1 and RRM2 domains shows minor spectral changes for both RRM1 and RRM2 (Fig. 1c and Supplementary Fig. 1a, c). In contrast, superposition of $^1$H-$^{15}$N HSQC spectra of RRM1-ZnF1$_S$ with RRM1-ZnF1$_S$-RRM2 indicates significant chemical shift differences for signals in the RRM1 and ZnF1 domains (Fig. 1d and Supplementary Fig. 1b, d). These differences could either result from a truncated construct or they could indicate transient interactions between RRM1 and ZnF1 that are stabilized in the tandem domain construct. To probe this further, we designed a new construct of the tandem domains, which includes the L1 linker (RRM1-ZnF1$_S$-L1, residues 94−230). This construct indeed shows only minor chemical shift differences when compared to RRM1-ZnF1$_S$-RRM2 (Fig. 1d and Supplementary Fig. 1b, d). These data suggest that the RRM1-ZnF1$_S$ construct might be affected by a truncated C-terminal boundary. Interestingly, average rotational correlation time ($\tau_c$) values (obtained from $^{15}$N $R_1$ and $R_{1\rho}$ experiments) for RRM1 and ZnF1 are comparable in both constructs, i.e. with and without linker L1 (Fig. 1e, f). Experimental $\tau_c$ values are in the order of theoretical $\tau_c \approx 8.4$ ns expected for a 14 kDa protein and significantly larger for the individual domains (expected $\tau_c \approx 6.5$ ns for 10.9 kDa RRM1 and $\tau_c \approx 1.9$ ns for the 3.2 kDa ZnF1). Taken together, these data are indicative of a strong coupling of the RRM1 and ZnF1 domains.

We crystallized the RRM1-ZnF1 construct and solved the structure to an overall resolution of 2.4 Å using Zn$^{2+}$-SAD phasing (Table 1 and Supplementary Fig. 2a). In the crystal structure, RRM1 adopts the canonical RRM fold with βαββαβ topology[30] and the ZnF1 adopts a canonical RanBP2-type zinc finger fold with two short β-hairpins sandwiching a conserved tryptophan and the Zn$^{2+}$ ion, which is coordinated by four cysteine residues[44]. The structure shows that the two domains interact via a hydrophobic interface involving the β-sheet of the RRM and the very C-terminal part of the construct. NMR and SAXS data show that the domain arrangement seen in the crystal structure

**Table 1 | Data collection and refinement statistics of crystal structures of RRM1-ZnF1 apo and the RRM1-ZnF1$_S$/RNA complex**

| Dataset | RRM1-ZnF1 PDB ID: 7PCV | RRM1-ZnF1$_S$ + GGCU_10 PDB ID: 7PDV |
|---|---|---|
| Wavelength (Å) | 0.87 | 1.28 |
| Resolution range (Å) | 32.1–2.42 (2.5–2.4)ᵃ | 45.1–3.5 (3.6–3.5) |
| Space group | C 1 2 1 | P 1 21 1 |
| Unit cell (Å) | 60.38 39.88 96.85 90 96.031 90 | 51.23 65.89 94.91 90 90.046 90 |
| Total reflections | 26,714 (2103) | 25,683 (4171) |
| Unique reflections | 8711 (751) | 7687 (774) |
| Multiplicity | 3.1 (2.8) | 1.8 (1.83) |
| Completeness (%) | 97.6 (85.8) | 94.1 (94.4) |
| Mean I/sigma(I) | 5.60 (0.8) | 1.98 (1.0) |
| Wilson B-factor | 52.36 | 53.96 |
| R-merge | 0.15 (1.20) | 0.36 (1.11) |
| R-meas | 0.18 (1.48) | 0.41 (1.13) |
| R-pim | 0.10 (0.84) | 0.23 (0.69) |
| CC1/2 | 0.99 (0.18) | 0.93 (0.52) |
| Reflections used in refinement | 8707 (751) | 7687 (774) |
| Reflections used for R-free | 436 (38) | 537 (57) |
| R-work | 0.26 (0.39) | 0.34 (0.32) |
| R-free | 0.31 (0.40) | 0.39 (0.43) |
| Number of non-hydrogen atoms | 1889 | 3426 |
| Macromolecules | 1850 | 3422 |
| Ligands | 2 | 4 |
| Solvent | 37 | 0 |
| Protein residues | 232 | 428 |
| RMS(bonds) (Å) | 0.003 | 0.002 |
| RMS (angles) (°) | 0.69 | 0.61 |
| Ramachandran favored (%) | 97.37 | 87.62 |
| Ramachandran allowed (%) | 2.19 | 12.14 |
| Ramachandran outliers (%) | 0.44 | 0.24 |
| Rotamer outliers (%) | 2.6 | 0.00 |
| Clashscore | 11.62 | 10.9 |
| Average B-factor | 58.16 | 31.47 |
| Macromolecules | 58.31 | 31.52 |
| Ligands | 46.86 | 25.04 |
| Solvent | 51.44 | – |
| Coordinate error (Å, Luzzati plot) | 0.44 | 1.0 |

ᵃStatistics for the highest-resolution shell are shown in parentheses.

**Table 2 | List of RNA oligonucleotides used in this study**

| RNA | RNA sequence |
|---|---|
| CU_9 | 5'–UCUCUUCUC–3' |
| GGCU_10 | 5'–UGGCUCUUCU–3' |
| GGCU_12 | 5'–UGGCUCUUCUCU–3' |
| eGGCU_12 | 5'–ACUUGGCUCUUCUCU–3' |

represents the solution conformation, thus excluding crystallization artifacts (Supplementary Fig. 2b–g and Supplementary Table 1). However, this arrangement of the tandem domains is not maintained in the three-domain construct (Fig. 1d). This argues that the arrangement of the RRM1-ZnF1 domains does not reflect the structure in the context of the triple domain construct (Fig. 1d and Supplementary Fig. 1d). Nevertheless, the RRM1 and ZnF1 domains are coupled together in solution in the longer constructs, as seen by comparable tumbling correlations times derived from $^{15}$N $R_1$ and $R_{1\rho}$ experiments (Fig. 1e, f). It is therefore not surprising that a comparison of $^1$H-$^{15}$N HSQC spectra of RRM1-ZnF1$_S$-RRM2 with the single RRM1 shows minor chemical shift differences all across RRM1 (Supplementary Fig. 1c). The RRM2 domain shows a slightly smaller tumbling correlation time in the three-domain construct and the L1 linker is highly flexible on the sub-nanosecond time scales, indicating that the tumbling of the second RRM is decoupled from the RRM1-ZnF1 module.

## RNA recognition by RBM5

RBM5 has been shown to recognize a pyrimidine-rich intronic region upstream of the ln100 regulatory sequence element for alternative splicing modulation of *caspase-2* pre-mRNA[21]. For RBM5 RRM2, a rather promiscuous RNA binding to GA or UC-rich RNAs has been reported with low micromolar affinities[38]. To evaluate the contributions of the different RNA binding domains of RBM5 we used an RNA ligand derived from *caspase-2* pre-mRNA - GGCU_12 (Table 2), which comprises a pyrimidine-rich region preceded by a GG dinucleotide and several protein constructs combining the different RNA binding motifs (RRM1-ZnF1$_S$-RRM2). NMR titration of GGCU_12 into the three RNA binding domains shows binding in fast to intermediate exchange on the NMR chemical shift timescale (Fig. 2a). The significant line-broadening observed reflects binding kinetics and an overall reduced tumbling of the complex. Notably, backbone amide resonances of all three domains show chemical shift changes indicating that all the RNA binding domains of RBM5 interact with the 12-residue GGCU_12 RNA and interestingly, significant chemical shift perturbations (CSPs) are also observed in the linker L1 indicating either a direct interaction of the linker with RNA or allosteric effects (Fig. 2a, b and Supplementary Fig. 3a). In addition, a 1:1 binding stoichiometry is observed by static light scattering (Supplementary Fig. 3b) measurements.

To analyze whether the RNA recognition is similar for single and tandem domains and ascertain if the linker L1 is involved in RNA binding, we compared CSPs seen for L1 extended versions of RRM1-ZnF1 (RRM1-ZnF1$_S$-L1) or RRM2 (L1-RRM2) upon binding to GGCU_12 RNA versus those seen for RRM1-ZnF1$_S$-RRM2 in the presence of GGCU_12 RNA (Fig. 2b, c and Supplementary Fig. 3a, c, d). The similarity of NMR spectral changes upon RNA binding between the three constructs indicates that the RNA binding mode is largely conserved (Fig. 2b, c and Supplementary Fig. 3a, c, d). The minor CSPs observed in L1 upon titration of GGCU_12 into L1-RRM2 (Fig. 2b, c) together with the flexibility of linker L1 in $^{15}$N-relaxation experiments of RRM1-ZnF1$_S$-RRM2 in presence of GGCU_12 RNA (Supplementary Fig. 3e) argue that L1 is not involved in direct RNA binding. This is not surprising considering that the linker is rather negatively charged (Fig. 1b). The CSPs in the linker L1 upon RNA titration likely arise from an allosteric effect caused by the domain rearrangement after the simultaneous binding of the flanking regions to the RNA molecule.

To assess if the tandem RRM1-ZnF1 construct, which presents a domain interaction involving the putative RNA recognition interface (Supplementary Fig. 2a), binds RNA similarly as the linker-extended version or the three-domain construct, we performed NMR titrations and compared chemical shifts in the RNA bound states. Notably, while signals for both RRM1 and ZnF1 show chemical shift differences in the unbound form when comparing RRM1-ZnF1$_S$ with RRM1-ZnF1$_S$-L1 and RRM1-ZnF1$_S$-RRM2, NMR spectra in the presence of RNA are highly similar (Fig. 2d and Supplementary Fig. 4). Therefore, we conclude that RNA binding by the tandem RRM1-ZnF1 domains resembles the corresponding contacts in RRM1-ZnF1$_S$-RRM2.

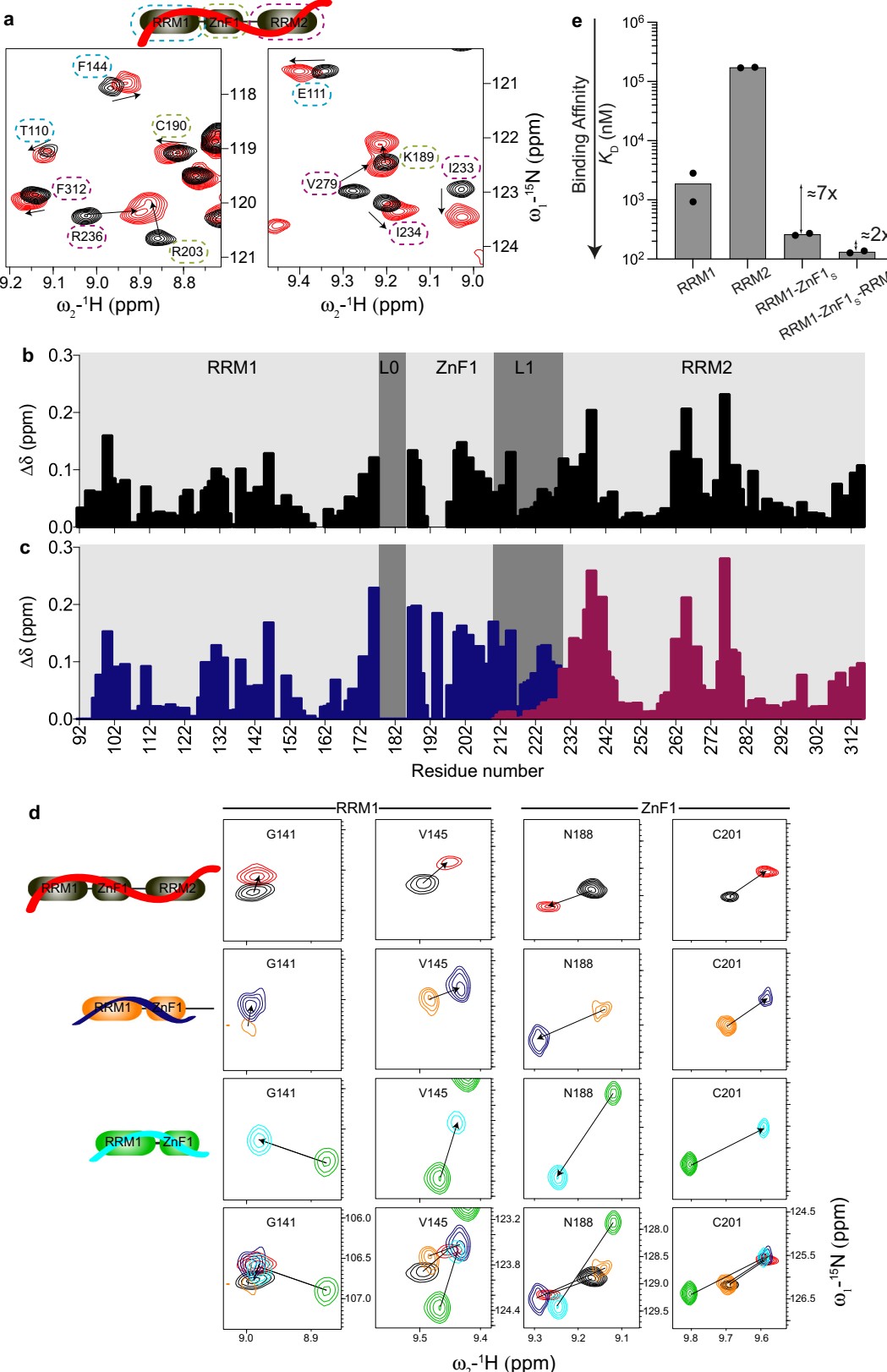

Finally, to dissect the contribution of the individual RNA binding domains to the overall RNA binding affinity we compared RNA binding of RRM1, RRM2, RRM1-ZnF1$_S$, and RRM1-ZnF1$_S$-RRM2 with GGCU_12 using ITC (Fig. 2e and Table 3). While RRM1 binds to GGCU_12 with a $K_D = 2$ μM, RRM2 binds ≈90-fold weaker with a $K_D = 173$ μM. The presence of ZnF1 improves the RNA binding affinity

7-fold (RRM1-ZnF1$_S$: $K_D = 263$ nM) while the affinity is only further increased 2-fold with the addition of the weaker binding RRM2 (RRM1-ZnF1$_S$-RRM2: $K_D = 133$ nM) (Fig. 2e and Supplementary Fig. 5a–d). While the increased affinity indicates some cooperativity in RNA binding, the effect is rather small, as the presence of the long-flexible linker L1 connecting RRM1-ZnF1 with RRM2 reduces the

**Fig. 2 | RNA binding by RRM1-ZnF1-RRM2 domains of RBM5 and individual domain contributions. a** Overlay of ${}^1$H-${}^{15}$N HSQC spectra of RRM1-ZnF1$_S$-RRM2 free (black) and bound to GGCU_12 RNA (red). Zoomed views of representative residues of RRM1 (blue), ZnF1 (green), and RRM2 (purple) are marked with dashed lines. **b** Chemical shift perturbations in RRM1-ZnF1$_S$-RRM2 (black) upon binding to GGCU_12 RNA at ratio of 1:1.1 (protein:RNA) vs. residue number are shown. **c** Chemical shift perturbations in RRM1-ZnF1$_S$-L1 (blue) and L1-RRM2 (maroon) upon binding to GGCU_12 RNA at ratios of 1:1.3 or 1:2 (protein:RNA) respectively vs. residue number are shown. **d** RNA binding by RRM-ZnF1$_S$-RRM2, RRM1-ZnF1$_S$-L1,

and RRM1-ZnF1$_S$ is similar. Zoomed views of ${}^1$H-${}^{15}$N HSQC spectra showing representative residues of RRM1 and ZnF1 of RRM-ZnF1$_S$-RRM2 -/+ GGCU_12 (black, red), RRM1-ZnF1$_S$-L1 -/+ GGCU_12 (orange, blue), and RRM1-ZnF1$_S$ -/+ GGCU_12 (green, cyan) are presented. **e** ITC-derived binding affinities of RRM1, RRM2, RRM1-ZnF1$_S$, and RRM1-ZnF1$_S$-RRM2 for GGCU_12 RNA show an increase in binding affinity with the addition of the individual domains. The bar plot shows the calculated dissociation constant ($K_D$) from an average of two measurements and the individual data points are shown as black dots. Source data are provided as a Source Data file.

## Table 3 | Isothermal titration calorimetry data for RBM5 RNA binding domains

| Protein | RNA | Av. $K_D{}^a$ (µM) | $K_D{}^b$ (µM) | ΔH$^b$ (kcal/mol) | −TΔS$^b$ (kcal/mol) | Binding stoichio-metry (N)$^b$ |
|---|---|---|---|---|---|---|
| RRM1 | CU_9 | 19.7 ± 6.5 | 13.2 ± 3.1 | −7.3 ± 0.8 | 0.61 | 1.04 |
| | | | 26.2 ± 6.4 | −10.0 ± 1.7 | 3.76 | 1 (fixed) |
| RRM1 | GGCU_12 | 1.88 ± 0.95 | 0.93 ± 0.18 | −11.7 ± 0.4 | 3.49 | 0.97 |
| | | | 2.83 ± 0.47 | −10.6 ± 0.4 | 3.06 | 1.01 |
| RRM2 | GGCU_12 | 172 ± 6 | 171 ± 8 | −24.5 ± 0.7 | 19.2 | 1 (fixed) |
| | | | 174 ± 8 | −25.8 ± 0.8 | 20.6 | 1 (fixed) |
| RRM1-ZnF1 wt | CU_9 | 5.56 ± 2.83 | 8.39 ± 0.87 | −11.8 ± 1.1 | 4.88 | 1.02 |
| | | | 2.73 ± 0.37 | −6.23 ± 0.27 | 1.36 | 0.94 |
| RRM1-ZnF1 wt | GGCU_12 | 0.153 ± 0.069 | 0.084 ± 0.012 | −16.2 ± 0.2 | 6.57 | 1.02 |
| | | | 0.222 ± 0.040 | −14.1 ± 0.3 | 4.99 | 0.81 |
| RRM1-ZnF1 F142/144A | GGCU_12 | 0.448 ± 0.036 | 0.469 ± 0.055 | −12.7 ± 0.2 | 4.05 | 0.7 |
| | | | 0.427 ± 0.047 | −12.0 ± 0.2 | 3.31 | 0.97 |
| RRM1-ZnF1 F202A | GGCU_12 | 0.379 ± 0.041 | 0.368 ± 0.046 | −9.58 ± 0.20 | 0.80 | 1.03 |
| | | | 0.390 ± 0.067 | −9.74 ± 0.27 | 1.00 | 1.01 |
| RRM1-ZnF1 R198E | GGCU_12 | 2.66 ± 0.38 | 2.69 ± 0.27 | −10.7 ± 0.3 | 3.08 | 1.05 |
| | | | 2.63 ± 0.70 | −8.37 ± 0.83 | 0.75 | 1.11 |
| RRM1-ZnF1$_S$ | GGCU_12 | 0.263 ± 0.027 | 0.273 ± 0.036 | −10.5 ± 0.2 | 1.56 | 1.08 |
| | | | 0.253 ± 0.041 | −10.9 ± 0.2 | 1.94 | 1.06 |
| RRM1-ZnF1$_S$ | eGGCU_12 | 0.641 ± 0.125 | 0.766 ± 0.116 | −15.2 ± 0.4 | 6.90 | 0.63 |
| | | | 0.516 ± 0.077 | −14.2 ± 0.3 | 5.66 | 0.60 |
| RRM1-ZnF1$_S$-RRM2 | GGCU_12 | 0.133 ± 0.012 | 0.127 ± 0.007 | −19.1 ± 0.1 | 9.68 | 0.98 |
| | | | 0.138 ± 0.022 | −18.1 ± 0.4 | 8.70 | 1.01 |

${}^a K_D$ values determined from duplicate measurements, with standard deviations as indicated.
${}^b$Values obtained for individual measurements are listed.

domain cooperativity and contribution of RRM2 to the overall affinity[45].

## Crystal structure of RRM1-ZnF1 in complex with RNA

To understand the details of RNA recognition by the tandem RRM1-ZnF1 domains at high resolution, we crystallized RRM1-ZnF1$_S$ with RNA. Crystals of RRM1-ZnF1$_S$ bound to GGCU_10 RNA (lacking two bases at the 3′-end of GGCU_12, Table 2) diffracted to an overall resolution of 3.5 Å. The structure was solved with molecular replacement using the RBM5 RRM1-ZnF1 apo structure (Table 1) and showed interpretable electron density for seven RNA nucleotides (UGGCUCU; Supplementary Fig. 6). The RNA is sandwiched between the RRM1 and ZnF1 tandem domains (Fig. 3a, b) and is recognized via a network of hydrophobic and polar interactions. The first uracil (U1) base forms hydrogen bonds with Arg140 from RRM1 and N2 of guanine at position 2 (G2) (Fig. 3c). G2 further stacks with Met101 and RNP1 aromatic residue Phe142 of RRM1. G3 is involved in an intricate set of specific interactions spanning both RRM1 and ZnF1 (Fig. 3d). It stacks with Phe202 of ZnF1 and backbone of Lys133, Met132 of RRM1, while its phosphate backbone is recognized by polar interactions with Asn194 and Arg198 of ZnF1 (Fig. 3d). Arg203 in ZnF1 forms hydrophobic contacts with C4 and Asp128 from the RRM1 establishes hydrogen bonds with the riboses of both C4 and U5 (Fig. 3e). U7 base is specifically recognized by the formation of a hydrogen bond between its 2′-oxygen and the hydroxyl group of Tyr148 from RRM1. In addition, the backbone carbonyl groups of Pro126 and Ala127 form a hydrogen bond

network with the 2′-hydroxyl and 3′-oxygen of the ribose sugar of U7, respectively. U5 and C6 are solvent exposed and not involved in any base-specific interactions. In summary, the specific recognition of the RNA core (U1-G2-G3) involves a hydrophobic groove between RRM1 and ZnF1 (Phe142, Met101, Met132, and Phe202) and important electrostatic interactions involving residues from both domains (Arg140, Asn194, and Arg198).

RBM5 is known to bind sequences containing a GG dinucleotide[10,23,37,39], which can be largely attributed to selective binding by its ZnF1 domain[37]. However, our crystal structure reflects an important contribution of RRM1 towards this specific recognition. We therefore probed the role of each domain for the recognition of the GG dinucleotide. Using isothermal titration calorimetry (ITC), we compared titrations of GGCU_12 RNA and CU_9 RNA which comprises a pyrimidine-rich region alone without the GG dinucleotide (Table 2), into RRM1 and the tandem RRM1-ZnF1 domains. We find that RRM1 binds to CU_9 with a $K_D ≈ 20$ µM (Supplementary Fig. 5e), a 10-fold lower affinity compared to GGCU_12 ($K_D ≈ 2$ µM, Supplementary Fig. 5a) (Table 3), indicating a significant contribution of RRM1 towards recognition of the GG dinucleotide. A modest 3.3-fold gain in affinity is achieved upon titration of CU_9 into tandem RRM1-ZnF1 domains ($K_D = 6$ µM, Supplementary Fig. 5f) compared to RRM1 ($K_D = 20$ µM, Supplementary Fig. 5e) suggesting that ZnF1 contributes somewhat towards recognition of the pyrimidine-rich CU_9 (Table 3). However, addition of the GG motif leads to a substantial 36-fold increased affinity by RRM1-ZnF1 ($K_D = 153$ nM, Supplementary Fig. 5g) when

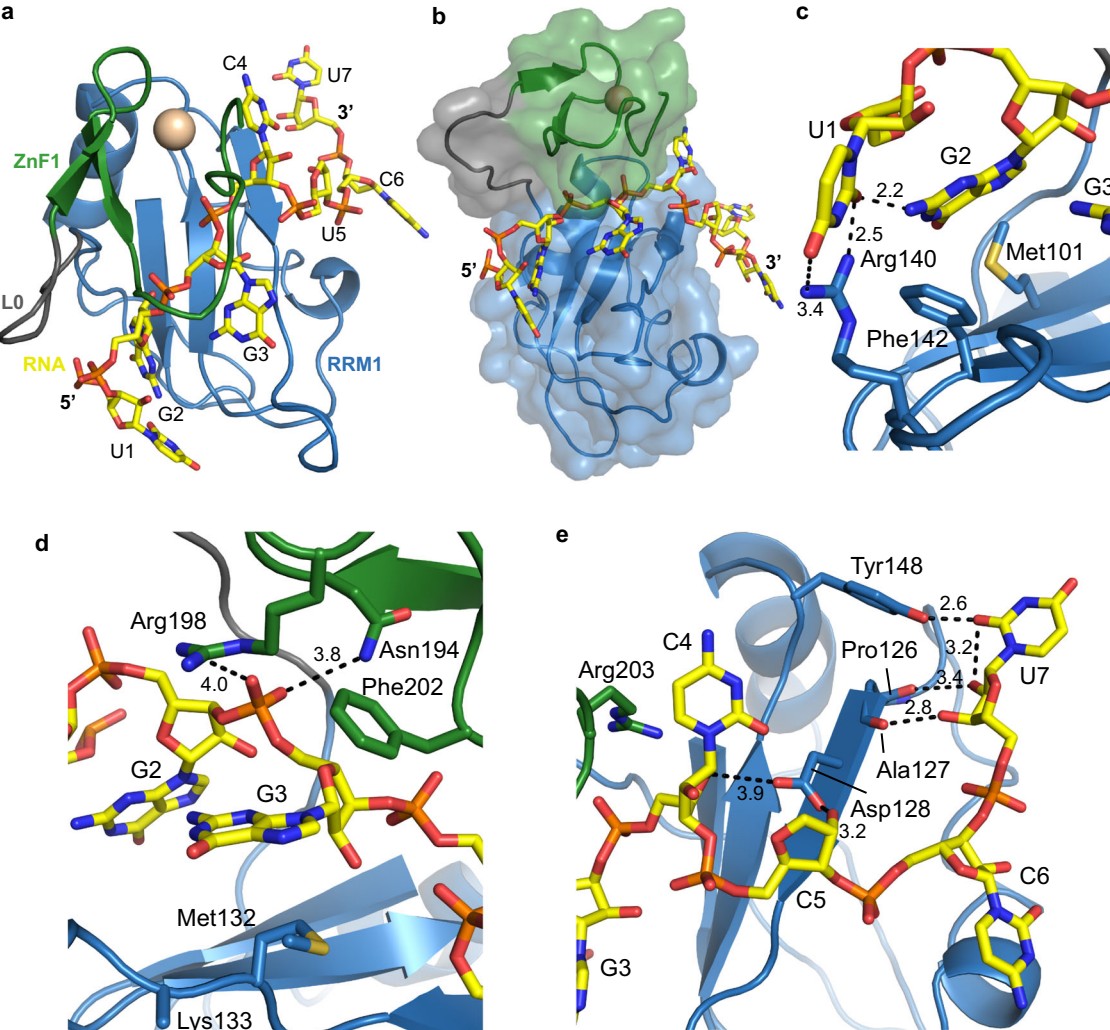

**Fig. 3 | Structural characterization of RRM1-ZnF1-RNA complex. a** Structure of RRM1-ZnF1$_S$ bound to GGCU_10 RNA in top view. RRM1, L0, ZnF1, and RNA are shown in blue, gray, green, and yellow respectively. The Zn²⁺ ion coordinated by the ZnF1 is shown in gold. **b** Surface representation of the tandem RRM1-ZnF1$_S$ domains. **c–e** Zoomed in views of the protein–RNA interface focusing on interactions between RRM1-ZnF1$_S$ and RNA nucleotides at positions (**c, d**) 1–3 and (**e**) 4–7, respectively. Hydrogen bonds are shown as dotted lines and distances are indicated in Å. See Supplementary Fig. 6 for $F_O$–$F_c$ omit maps.

compared to the pyrimidine-rich CU_9 RNA alone ($K_D$ = 6 μM) or a 12-fold increase in affinity compared to the RRM1 construct. In conclusion, these ITC experiments clearly show that both RRM1 and ZnF1 are important for recognition of the GG dinucleotide in GGCU_12 (Table 3). More generally, the crystal structure shows extensive hydrophobic and polar contacts with the GG dinucleotide-containing RNA with both RRM1 and ZnF1 domains contributing to the overall affinity and specificity of the interaction.

**Mutational analyses of RRM1-ZnF1 for RNA binding and splicing**
Our structure of the RRM1-ZnF1 RNA complex shows that several hydrophobic and polar residues of RRM1 and ZnF1 contribute towards RNA binding. To probe their relative contributions, first ITC titrations were made in several single/double point mutants of the tandem RRM1-ZnF1 domains using GGCU_12 RNA (Table 3). Mutation in the RNP1 residues of RRM1 (F142A/F144A) in the context of tandem RRM1-ZnF1 domains leads to a ≈ 3-fold loss in affinity ($K_D$ = 448 nM; Fig. 4a and Supplementary Fig. 5h). In fact, the isolated RRM1 containing these RNP1 mutations completely loses its RNA binding capacity, as shown by NMR titrations where no chemical shift perturbations are observed upon addition of CU_9 into ¹⁵N-labelled F142A/F144A RRM1 (Fig. 4b). In addition, aromatic (F202A) and positively charged residues (R198E) in

ZnF1 were changed to assess their respective contribution to RNA binding. While the F202A mutation leads to only a ≈ 2.5-fold loss in affinity ($K_D$ = 379 nM, Fig. 4a and Supplementary Fig. 5i), the charge reversal mutation (R198E) shows a dramatic (18-fold) reduction in affinity ($K_D$ = 2.7 μM; Fig. 4a and Supplementary Fig. 5j). Since R198 is involved in polar interactions with the backbone phosphate of G3 of the RNA (Fig. 3d), a charge reversal mutation would not only inhibit the interaction but would also cause electrostatic repulsions leading to the observed pronounced decrease in affinity.

To assess the functional importance of the RNA recognition by RRM1 and ZnF1 for splicing regulation we performed minigene reporter assays comparing wild-type protein and two double mutants in RRM1 (F142A/F144A, referred as FAFA) or ZnF1 (K197E/R198E, referred as KERE), which strongly impair RNA binding. We first compared their splicing activities to that of the wild-type protein using assays in which expression vectors of these proteins were co-transfected in HeLa or HEK 293 cells with a minigene reporter. While overexpression of RBM5 affected AS of *Caspase-2* pre-mRNA in a *Casp-2* minigene reporter system[21,22], the response generated in these assays was not sufficiently robust to allow quantification of the effects of point mutations. As an alternative, we analyzed the effects of RBM5 overexpression on *NUMB* exon 9 AS, which is a key target of RBM5, 6

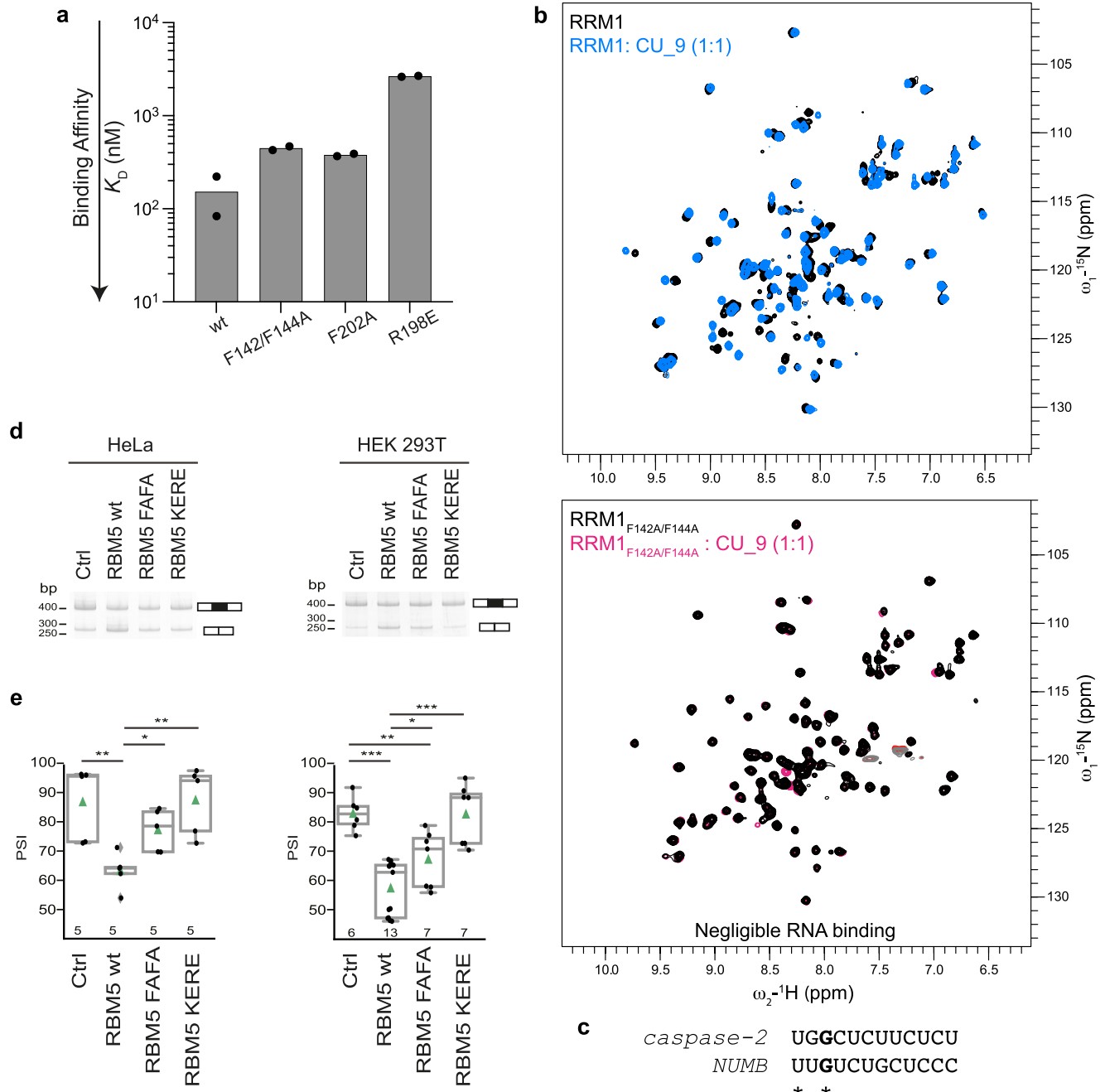

**Fig. 4 | Mutational analysis of RRM1-ZnF1. a** RNA binding contribution of specific residues of RRM1 and ZnF1, as probed by measuring binding affinity of RRM1-ZnF1 point mutants to GGCU_12 RNA using ITC. The bar plot shows the calculated dissociation constant ($K_D$) from an average of two measurements and the individual data points are shown as black dots. **b** A superposition of $^1$H-$^{15}$N HSQC spectra of RRM1 wild-type in free (black) and in presence of CU_9 RNA (sky blue) at a protein:RNA ratio of 1:1 is shown in the upper panel. A double mutant in RRM1 RNP1 residues (F142A/F144A) does not bind RNA as seen by a superposition of $^1$H-$^{15}$N HSQC spectra of RRM1$_{F142A/F144A}$ mutant in free (black) and in presence of CU_9 RNA (pink) at a protein:RNA ratio of 1:1 in the lower panel. **c** Sequence alignment of *caspase-2* derived RNA sequence used for in vitro studies and the RBM10 binding CLIP-Seq consensus motifs used for the construction of *NUMB* minigene reporter[23]. (*) indicates identity and (.) indicates similarity due to a pyrimidine to pyrimidine substitution. **d** HeLa and HEK 293T cells were co-transfected with *RG6-NUMB* alternative splicing reporter[23] and T7-RBM5 vectors expressing wild type or RNA binding affinity mutants in the RRM1 (F142A/F144A->FAFA) or ZnF1 (K197E/R197E->KERE) or control vector, as indicated. Pattern of alternative splicing isoforms was detected by RT-PCR using primers complementary to vector sequences flanking exons of the *RG6-NUMB* minigene; the positions of the amplification products corresponding to exon 9 inclusion/skipping are indicated. The results correspond to one representative replicate of the experiment. Uncropped gels are provided in Source Data. **e** Quantifications of alternatively spliced isoforms shown in panel **d** for the number of biological replicate experiments indicated at the bottom of each bar were used to generate the boxplots shown (bottom panel). The box represents the interquartile range from the 25th percentile to the 75th percentile, the median is represented by the line in the box. The whiskers represent the minimal and maximal values and the outliers are plotted as gray diamonds. The means are indicated with the green triangles, each black dot represents a biological replicate. T-test (two-tailed distribution, homoscedastic) results are indicated (*<0.05, **<0.01; ***<0.001). Source data are provided as a Source Data file.

and 10 in the regulation of cancer cell proliferation[23]. We used a *NUMB* exon 9 reporter that was previously shown to respond to over-expression of RBM6 and RBM10 proteins[23]. *NUMB* exon 9 includes a GU motif followed by a pyrimidine-rich sequence that resembles the RBM5-responsive motif in *Casp-2*[21] (Fig. 4c), including a conserved G at position 3, which in our crystal structure of RRM1-ZnF1-RNA complex is recognized by both the RRM1 and ZnF1 (Fig. 3d). While overexpression of wild type RBM5 reduced the levels of *NUMB* exon 9 inclusion compared to control cells (a reduction of ≈20 Percent Spliced In (PSI) units) both in HeLa and in HEK 293T cells, equivalent levels of expression of either of the mutants resulted in more limited (FAFA) or negligible (KERE) effects (Fig. 4d, e and Supplementary Fig. 7). The particularly strong effects of the KERE mutation are consistent with the strong effects of this mutant in in vitro RNA binding experiments (Table 3). We conclude that key residues important for RNA binding also affect the functional activity of RBM5 in modulating alternative splicing.

### A structural model of the RRM1-ZnF1-RRM2 -RNA complex

We finally derived a structural model of the RRM1-ZnF1-RRM2/RNA complex by combining SAXS and NMR data together with the available structural information for RRM1-ZnF1 and RRM2 domains (Supplementary Fig. 8a, b and Supplementary Table 1), following previously described protocols[46,47]. The SAXS-derived pairwise distance distribution, p(r), of the three RRM1-ZnF1$_S$-RRM2 domains in the absence and presence of GGCU_12 shows significant compaction upon RNA binding and a reduction of the maximum pairwise distance $D_{max}$ of the particle ($D_{max}$ = 78 Å for RRM1-ZnF1$_S$-RRM2 apo and 75 Å for the RNA complex) (Fig. 5a and Supplementary Fig. 8c–f). Our NMR and ITC data show that all three domains (RRM1, ZnF1, and RRM2) can simultaneously bind to

GGCU_12 RNA (Fig. 2a, b, e). Taken together with the fact that the crystal structure of the RRM1-ZnF1/RNA complex shows only the first 7 residues of the GGCU_12 RNA being recognized by the RRM1-ZnF1 module (Fig. 3a), it is plausible that additional contacts with down-stream nucleotides in the 12 residue GGCU_12 RNA are mediated by RRM2.

Next, we measured long-range domain/domain distances using NMR paramagnetic relaxation enhancement (PRE) experiments. For this, several single cysteine mutants were generated in the different RBM5 domains and conjugated to single nitroxyl spin labels. Spin labeling at a total of four different positions was successful (Cys155 in RRM1 and Cys288, Cys249 and Cys307 in RRM2) and gave rise to PRE effects in the absence and presence of RNA (Supplementary Fig. 9a). While spin labels at Cys155 and Cys249 do not show inter-domain effects in both the apo and RNA-bound forms, spin labels at Cys288 and Cys307 show significant PRE effects. Cys288, which is located in helix α2 of RRM2, shows similar PRE effects for residues in the linker L1 and RRM2 in the absence and presence of RNA. Interestingly, a spin label at residue 307 (located in strand β4 of RRM2) shows significantly stronger PRE effects for amides in RRM1, ZnF1, and the linker L1 in the RNA complex compared to the unbound protein, indicating significant structural rearrangements.

To derive a structural model for the protein–RNA complex we generated 2000 structure models of the RBM5 three domains-RNA complex by randomizing the L1 linker. The RRM1-ZnF1 and RRM2 modules were treated as semi-rigid bodies, using our crystal structure of the RRM1-ZnF1-RNA complex and the reported NMR structure of RRM2 (PDB ID: 2LKZ) as templates. Together with PRE-derived distance restraints, chemical shift perturbations observed in RRM2 upon RNA binding were also incorporated as ambiguous RNA/RRM2

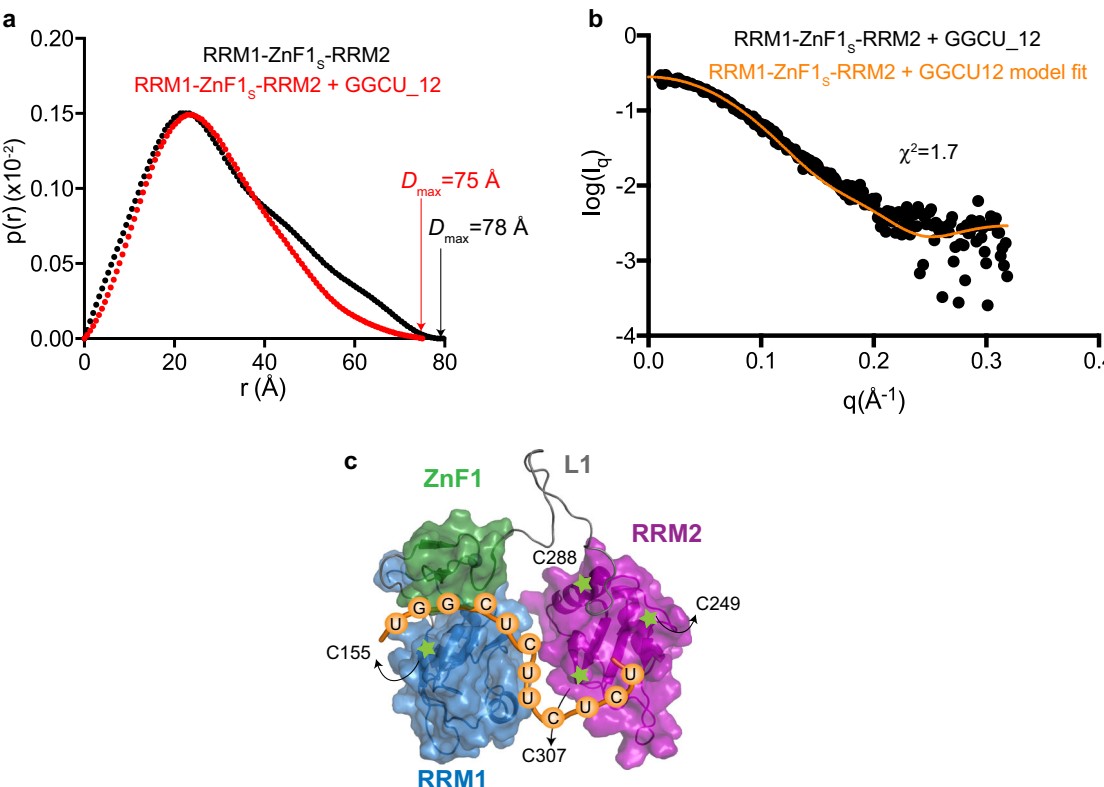

**Fig. 5 | Domain arrangement of RRM1-ZnF1-RRM2. a** Comparison of SAXS p(r) curves showing maximum pairwise distribution of RRM1-ZnF1$_S$-RRM2 free (black) and in complex with GGCU_12 RNA (red). $D_{max}$ is indicated for the respective SAXS curves. **b** The fit between experimental SAXS data for RRM1-ZnF1$_S$-RRM2 bound to GGCU_12 RNA against the simulated data from the top PRE-based model is plotted, as obtained using Crysol software, the $χ^2$ value is indicated. **c** Structural model of RRM1-ZnF1$_S$-RRM2 in the presence of RNA, as calculated based on PRE and SAXS data. Positions of spin labels are marked and the domains are shown in surface representation. Source data are provided as a Source Data file.

distance restraints during structure calculations. The structure calculation converged well, with the 10 lowest energy structures showing a backbone $C_\alpha$ coordinate RMSD of 1.52 ± 0.8 Å (Supplementary Fig. 9b). Notably, these structures are also in excellent agreement with the experimental SAXS data, with $\chi^2$ = 1.3–2.4. Analysis of the lowest energy structure model (SAXS $\chi^2$ = 1.7, Fig. 5b) reveals a compact arrangement with all three RNA binding domains contacting the RNA (Fig. 5c). Considering the RNA binding contributions of the individual domains based on our ITC (Fig. 2e) and SAXS data (Fig. 5a), this demonstrates that the RRM1, ZnF1, and RRM2 domains adopt a compact domain arrangement which is stabilized by cooperative contributions of all three domains for high-affinity RNA binding, even if the contributions by RRM2 are weaker and less sequence-specific.

## Discussion

In this study, we investigated the RNA binding properties of RBM5 to provide insights into its complex modes of target pre-mRNA recognition involving distinct but synergistic contributions by the RNA binding domains. We present the crystal structure of RBM5 RRM1-ZnF1 tandem domains in complex with a *caspase-2* derived pre-mRNA sequence. Using our RRM1-ZnF1-RNA crystal structure and solution NMR structure of apo-RRM2[38], together with our NMR-based PRE and SAXS experiments, we present a model of RNA recognition where all three RNA binding domains of RBM5 contribute to the overall affinity. ITC and NMR data provide details of a cooperative mode of action of the RNA binding domains of RBM5.

The crystal structure of the unbound RRM1-ZnF1 tandem domains shows an unexpected conformation where the two domains are coupled together (Supplementary Fig. 2). While this structure might not represent the "true" or average conformation of the tandem domains in the context of the three domains, NMR-relaxation data on the linker-extended and three-domains constructs nevertheless suggest a certain degree of coupling between the two domains in solution (Fig. 1e, f). Surprisingly, this is in stark contrast to the closely related protein RBM10 where RRM1 and ZnF1 domains in the tandem RRM1-ZnF1 construct tumble completely independent of each other in solution[40], even though the length and sequence of linker L0 is rather similar between the two proteins (Fig. 1b). The reason for such differences in the coupling of the RRM1 and ZnF1 domains between RBM5 and RBM10 likely stems from variations at the C-terminal part of the ZnF1. While RBM5 harbors a phenylalanine at position 209 that stacks on the hydrophobic core of RRM1 stabilizing the inter-domain interface, RBM10 contains a glutamic acid (Fig. 1b and Supplementary Fig. 2a). In the context of the three domains, the slightly reduced correlation time of residues in RRM2 compared to the RRM1-ZnF1 module indicates partial decoupling of RRM2, likely reflecting the flexibility of the linker L1 in RBM5. Collins et al. made similar observations with RBM10 although the differences in domain correlation times were much larger between the individual domains compared to that of RBM5 (9 ns, 8 ns, and 11 ns vs. 13 ns, 13 ns, and 12 ns for RRM1, ZnF1, and RRM2 in RBM10 and RBM5, respectively). It is noteworthy however that the linker L1 significantly differs in length between RBM5 (20 residues) and RBM10 (50 residues) suggesting a much greater degree of conformational decoupling of RRM2 in the latter[40].

Previous studies focusing on RanBP2-type zinc fingers (including RBM5) indicate a binding specificity for ssRNAs containing GGU sequence motifs with micromolar affinities[37,48]. Indeed, the RBM10 tandem RRM1-ZnF1 domains bind to a CLIP motif CUGU<u>GG</u>A with $K_D$ = 74 nM and mutation of either guanosine of the GG dinucleotide leads to a 14–16 fold decrease in affinity[40]. A lower affinity interaction has been reported for RBM10 RRM1-ZnF1 and ZnF1 with a GG containing 22-nucleotide sequence, derived from *Fas* pre-mRNA, with $K_D$ = 412 nM and 845 nM, respectively[49]. To some extent affinity differences might reflect variations in binding assays used (ITC vs. membrane filter binding assays). Notably, our data show that the GG

dinucleotide in *caspase-2* pre-mRNA is recognized by both the RRM1 and ZnF1 domains of RBM5, as seen by comparing binding affinities of RRM1 alone and tandem RRM1-ZnF1 domains to the pyrimidine-rich CU_9 RNA and the GGCU_12 RNA (Table 3).

Surprisingly, our crystal structure of RRM1-ZnF1$_S$ tandem domains in complex with a *caspase-2* derived RNA (GGCU_10), containing both a GG dinucleotide and a pyrimidine-rich sequence shows that the RNA is sandwiched between the two domains with specific RNA recognition mediated by both the RRM1 and ZnF1 domains. We show that RRM1 and ZnF1 residues important for RNA binding also impact the activity of RBM5 in the NUMB exon 9 reporter assays. Note, that the in vitro studies focus only on the RNA binding domains, while the splicing assays are performed with full-length protein, where additional domains present may contribute to splicing activity via protein-protein interactions.

While U1 is specifically recognized by Arg140 of RRM1 and G3 is recognized by hydrophobic contacts with Lys133, Met132 of RRM1, Phe202 of ZnF1 and polar interactions with Asn194 and Arg198 of ZnF1, no interactions of the ZnF1 are observed with G2. In the ZRANB2 F2 zinc finger, the GG dinucleotide is recognized by stacking of a unique guanosine-Trp-guanosine ladder which is further supported by hydrogen bonds from Asn76, Asn86, Arg81, and Arg82 (Supplementary Fig. 10a). In RBM5, a phenylalanine (Phe195) is present instead of the central tryptophan, and Asn76/Asn86 are replaced by Leu192/Phe202 (Supplementary Fig. 10b). Unfortunately, electron density for Phe195 side-chain is not visible in our crystal structure and Lys197 (corresponding to Arg81 in ZRANB2 F2) is solvent exposed (Supplementary Fig. 10b). Notably, RRM1 and RRM2 both lack a canonical aromatic residue in RNP2 (Met101 in RRM1 and Ile234 in RRM2; Fig. 1b and Supplementary Fig. 10c), and therefore the major RNA interaction comes from RNP1 as mutation of its aromatic residues completely abolishes RNA binding by RRM1 (F142/144A, Fig. 4b) or significantly decreases RNA binding by RRM2 (F276A, F278A)[38]. In the canonical RRM-RNA interaction, the aromatic residues at positions 2 and 5 of RNP2 and RNP1 stack with two adjacent nucleotides in the RNA, while the aromatic residue at position 3 of RNP1 is usually inserted between the sugar rings of this RNA dinucleotide (Supplementary Fig. 10d and [50]). The RNA is involved in hydrophobic interactions with position 2 of RNP2 even in the absence of an aromatic residue (Supplementary Fig. 10d). In our structure however, no hydrophobic interactions are observed between the RNA and the aromatic residue at position 5 of RNP1 while position 3 of RNP1 (Phe142) and position 2 of RNP2 (Met101) stack with G2 of RNA in a non-canonical way (Supplementary Fig. 10d). Taken together, our data show a non-canonical mode of RNA recognition by RRM1 and ZnF1, which stems from a combination of non-canonical RNA-interfacing residues in both RRM1 and ZnF1 and conformational restrictions posed by sandwiching of the RNA between the tandem domains. Notably, the sequence changes driving this non-canonical mode of RNA recognition by RRM1-ZnF1 domains in RBM5 are remarkably similar to those observed in RBM10 as well (Fig. 1b). It therefore seems likely that RBM10 RRM1 and ZnF1 domains also recognize RNA harboring a GG dinucleotide non-canonically. Interestingly, in RBM10, two nucleotides upstream of the GG dinucleotide also contribute to RNA binding[40]. To test if this is also the case for RBM5, we titrated a 5′-extended variant of the GGCU_12 RNA (eGGCU_12, Table 2) harboring three additional nucleotides based on *caspase-2* pre-mRNA. In contrast to RBM10, we find that eGGCU_12 binds 2.4-fold weaker to RBM5 RRM-ZnF1$_S$ when compared with GGCU_12 ($K_D$ = 6 μM, Supplementary Fig. 5k). The binding stoichiometry is also altered, which suggests a second suboptimal binding event due to an increase in the RNA length (Table 3).

Collins et al. suggested that RBM10 RRM2 preferentially binds C-rich sequences with a moderate low micromolar affinity[40]. This is in contrast to RBM5 RRM2, which has rather broad and promiscuous

RNA binding preference and can recognize CU- and GA-rich RNA sequences with similar affinities[38]. The RNA ligand used for our structural studies represents a combination of RNA sequence motifs known to bind ZnF1 (GG)[37] and RRM2 (C/U rich)[38]. However, RBM5 RRM2 on its own binds GGCU_12 with a rather weak, high micromolar affinity ($K_{D\ RRM1}$ = 173 μM, Table 3) and its addition to the RRM1-ZnF1 module in the three domains construct also only leads to a modest 2-fold gain in affinity for GGCU_12 (Table 3). These data indicate that RRM2 alone might not contribute to specificity in recognition and regulation of *caspase-2* and *NUMB* pre-mRNA sequences. Nevertheless, a point mutation in RRM2 (R263P) causes pre-mRNA splicing defects in round spermatids affecting male fertility in mice[51]. A superposition of $^{1}$H,$^{15}$N-HSQC NMR spectra of the RRM2 R263P mutant and wild-type proteins shows severe protein folding defects in the mutant protein (Supplementary Fig. 11). This suggests that structural disturbances in the RBM5 RRM2 domain, possibly affecting both RNA binding and protein-protein interactions, directly translate into functional defects. Given the low affinity contribution of RRM2 towards RNA we observe, it is likely that either the RNA sequences studied here do not satisfy the binding preferences of RRM2 or that RRM2 indirectly affects splicing regulation by interacting with *trans*-acting protein factors.

Our ITC data show that the combination of three RNA binding domains indeed leads to an increase in the overall binding affinity (RRM1 → RRM1-ZnF1$_S$ → RRM1-ZnF1$_S$-RRM2: 2μM → 263 nM → 133 nM), indicating cooperative binding of the domains. Cooperativity in multidomain proteins is common and has been found for many multidomain RNA binding proteins[45,52-56]. While the individual RRM domains (especially the RBM5 RRMs, which even lack a canonical aromatic residue) have limited capacity to recognize RNA in a sequence-specific manner, their combination with additional domains can increase sequence specificity, and at the same time strengthen RNA binding affinity by creating a larger binding interface[30]. The relatively short linker L0 and the coupling between RRM1 and ZnF1 suggest that RRM1-ZnF1 acts as an independent module for specific RNA recognition. RRM2 acts as an independent RNA binding module as well, where the flexible linker L1 may enable local diffusion and initial RNA scanning by RRM2.

The role of RBM5 in the regulation of alternative splicing seems to be complex as it is able to regulate distinct processes in a cell-type specific manner. Therefore, it seems likely that RBM5 uses different modules to recognize and access various RNA ligands based on the sequence motifs presented to it. Our structural analysis of the three RBM5 RNA binding domains shows that the protein adopts a compact conformation upon RNA binding (reduction in maximum dimensions of the protein upon RNA binding from 78 Å to 75 Å, Fig. 5a), even though the L1 linker remains flexible (Supplementary Fig. 3e). The length and flexibility of the L1 linker may play a role and enable initial scanning of an RNA and may also modulate RNA binding, as has been observed for other multidomain RNA binding proteins with long, flexible linkers connecting the individual domains. Here, transient interactions of the linker with flanking RNA binding domains can modulate RNA binding affinity and specificity[46,53,57,58]. Future studies should focus on providing high-resolution details for the contributions of all three domains for the recognition of extended RNA ligands and the specific roles of the L1 linker and RRM2.

The present study provides unprecedented molecular and structural insights into pre-mRNA target binding and regulation involving the three RNA binding domains of RBM5 and reveals surprising differences to its close homolog RBM10. Our results show that RRM1-ZnF1 acts as a single moiety in a non-canonical mode of RNA recognition, RRM2 is more uncoupled in the absence of RNA while adopting a more rigid architecture upon RNA binding. Therefore, in the case of *caspase-2* pre-mRNA recognition, the RRM1-ZnF1 domains in the context of the complete N-terminus of RBM5 could act as a tether by specific

recognition of a GG dinucleotide with high nanomolar affinity and RRM2 could act as a helper domain to increase the overall binding affinity.

## Methods

### Protein expression and preparation

The DNA sequence encoding various constructs of RBM5 including RRM1 (residues 94-184), RRM2 (residues 231-315) and RRM1-ZnF1-RRM2 (residues 94-315) were cloned into modified pET24d vector containing an N-terminal His-Thioredoxin tag upstream of a TEV protease cleavage site, while RRM1-ZnF1 (residues 94-210) was cloned into pET28a vector. All point mutations were generated using overlapping PCR and the results were confirmed by Sanger sequencing. The primers used to generate all constructs in this study are listed in Supplementary Table 2. All plasmids were then transformed into chemically competent *E.coli* BL21 (DE3) cells for expression. To produce unlabeled, $^{15}$N- or $^{13}$C-$^{15}$N uniformly labeled protein samples, the cells were grown either in Luria-Bertani (LB) or M9 minimal media supplemented with $^{15}$NH$_4$Cl/$^{13}$C-glucose and 50 μg/ml Kanamycin. For constructs containing the zinc finger (ZnF1) domain, the cultures were supplemented with 100 μM ZnCl$_2$ solution for proper folding of the domain. The protein was expressed by growing cultures to an optical density (OD) of 0.6–0.8 at 37 °C before induction with 0.5 mM IPTG and grown overnight at 18 °C (RRM1-ZnF1$_S$-RRM2) or 20 °C (RRM1, RRM2). For RRM1-ZnF1, RRM1-ZnF1$_S$, RRM1-GGS-ZnF1, and RRM1-ZnF1$_S$-L1, the cultures were grown for 3 h at 37 °C after induction. The cells were harvested by centrifugation and resuspended in buffer containing 20 mM Tris pH 7.5, 500 mM NaCl, 10 mM Imidazole (RRM1, RRM2) or 20 mM Hepes-Na, pH 7.5, 500 mM NaCl, 1 M Urea (RRM1-ZnF1, RRM1-ZnF1$_S$, RRM1-GGS-ZnF1 and RRM1-ZnF1$_S$-L1) or 20 mM sodium phosphate pH 7.0, 500 mM NaCl, 1 M Urea (RRM1-ZnF1$_S$-RRM2). All buffers were supplemented with 2 mM β-Mercaptoethanol (BME) and 1 mM AEBSF protease inhibitor. Subsequently, cells were lysed using sonication and lysates were centrifuged at 35000 × *g* for 45 min to separate cell debris and supernatant. For RRM2, the supernatant was loaded onto a 3 ml bench top Ni$^{2+}$ affinity column equilibrated with lysis buffer, washed with 30 ml of lysis buffer and eluted in the same buffer supplemented with 500 mM Imidazole. The eluted protein was incubated with TEV protease overnight at 4 °C for cleavage of fusion tag during dialysis to remove excess Imidazole. The protein was loaded again onto Ni$^{2+}$ affinity column to remove uncleaved protein and the fusion tag from cleaved protein of interest and the flow-through was collected. The protein was further purified over a size exclusion chromatography column (Hiload 16/60 Superdex75 column, GE Healthcare) equilibrated with buffer containing 20 mM MES pH 6.5, 100 mM NaCl, 1 mM DTT. For RRM1, a cation exchange purification step (1 ml Resource S column, GE Healthcare) was introduced before size exclusion chromatography, where the protein was diluted to contain 50 mM NaCl, loaded on the column and eluted with a linear gradient of buffer containing 20 mM Tris pH 7, 1 M NaCl. For RRM1-ZnF1, RRM1-ZnF1$_S$, RRM1-GGS-ZnF1, and RRM1-ZnF1$_S$-L1, the supernatant was diluted 5-fold to contain 100 mM NaCl and purified over SP-Sepharose (HiPrep SPFF, GE Healthcare) using a linear gradient of buffer containing 2 M NaCl. The eluted protein was diluted 3-fold to contain 10 mM potassium phosphate, pH 7.4, 75 mM NaCl, loaded on a homemade 15 ml hydroxyapatite (HA) column and eluted with a 2-step gradient of buffer containing 12% w/v (NH$_4$)$_2$SO$_4$. Finally, the protein was purified over a size exclusion column in buffer containing 20 mM MES pH 6.5, 400 mM NaCl, 1 mM DTT. For RRM1-ZnF1$_S$-RRM2, the supernatant was loaded on a 3 ml Zn$^{2+}$ affinity bench top column, equilibrated with lysis buffer, washed with buffer containing 1 M NaCl for removing non-specifically bound nucleic acids and sequentially eluted with buffer with pH adjusted to 6.0, 5.5, 5.0, 4.5. The protein appeared to be mostly pure in fractions with pH 5.5-4.5. TEV cleavage was done in 10 mM Na phosphate pH 7, 400 mM NaCl, 2 mM BME

buffer. The protein was diluted 8-fold to contain 50 mM NaCl and purified over a 6 ml Resource S column with a linear gradient from 50 mM NaCl to 1 M NaCl. The eluted peak fractions were concentrated to 1 ml protein solution, after adjusting the final salt concentration to 400 mM NaCl and finally purified over a size exclusion column with buffer containing 20 mM MES pH 6.5, 400 mM NaCl, and 1 mM DTT. The purity of final protein samples was checked on Coomassie-blue stained sodium dodecyl sulfate polyacrylamide gels (SDS-PAGE). The final concentrations of RBM5 protein constructs were calculated using NanoDrop 1000 spectrophotometer (Thermo Scientific) with absorbance measured at 280 nm using extinction coefficients of 18450 $M^{-1}cm^{-1}$, 16960 $M^{-1}cm^{-1}$, 13980 $M^{-1}cm^{-1}$, 8480 $M^{-1}cm^{-1}$, and 1490 $M^{-1}cm^{-1}$ for RRM1-ZnF1$_S$-RRM2, RRM1-ZnF1$_S$-L1, RRM1-ZnF1$_S$, RRM1, and RRM2, respectively.

### RNA synthesis
RNA oligonucleotides for crystallization, NMR and ITC were purchased from IBA GmBH, Germany. The RNA concentrations were calculated using NanoDrop 1000 spectrophotometer (Thermo Scientific) with absorbance measured at 260 nm using extinction coefficients of 75800 $L\,mol^{-1}\,cm^{-1}$, 87300 $L\,mol^{-1}\,cm^{-1}$, 103600 $L\,mol^{-1}\,cm^{-1}$ and 133400 $L\,mol^{-1}\,cm^{-1}$ for CU_9, GGCU_10, GGCU_12, and eGGCU_12, respectively.

### NMR spectroscopy
All spectra were recorded at 298 K on AVIII 600, AVIII800, or AVIII + 950 Bruker NMR spectrometers equipped with cryogenic triple resonance gradient probes using Topspin v3.2. Samples contained 0.05–0.6 mM protein in 20 mM MES pH 6.5, 100 mM (RRM1, RRM2)/ 400 mM (RRM1-ZnF1, RRM1-ZnF1-RRM2) NaCl, 1 mM DTT supplemented with 10 % $D_2O$ for lock. Spectra were processed in NMRPipe/Draw[59] and analyzed in CCPN Analysis[60]. Protein backbone resonance assignments were obtained using 3D HNCA, HNCACB, CBCA(CO)NH and HNCO[61] and side-chain resonance assignments using HCCH-TOCSY and H(CCO)NH[62]. Aromatic resonances were assigned using 2-D $^1H$–$^{13}C$ HSQC, HBCBCGCDHD and HBCBCGCDCEHE experiments[63]. Additionally, to check for presence of inter-molecular NOEs in the protein–RNA complex, a 2D ω$_1$-filtered NOESY, aliphatic and aromatic 3D ω$_1$-filtered edited $^{13}C$ NOESY experiments in 100% $D_2O$ were recorded at protein: RNA ratio of 0.8:1. To see the dispersion of RNA signals in free versus protein bound form, 2D $^1H$-$^1H$ TOCSY spectra were recorded. The chemical shift assignments for ZnF1 (ID:17387) and RRM2 (ID:18017) from the BMRB repository were used to assist in the assignment process, wherever necessary.

Since RBM5 RRM1-ZnF1$_S$, RRM1-ZnF1$_S$-L1, and RRM1-ZnF1$_S$-RRM2 were not stable at high concentrations at 100 mM NaCl in free form, the respective RNA was added to the protein at 1:1 ratio at 400 mM NaCl and subsequently diluted to 100 mM NaCl.

**NMR relaxation measurements.** Experiments to measure $^{15}N$ $R_1$ and $R_{1\rho}$ relaxation were performed[64,65] to study the dynamic properties of the different protein constructs. NMR data were recorded at 298 K for $^{15}N$-uniformly labeled proteins at 0.24 mM concentration for wild-type RRM1-ZnF1, 0.57 mM for RRM1-ZnF1$_S$, 0.16 mM for RRM1-GGS-ZnF1, 0.20 mM for RRM1-ZnF1$_S$-L1 and 0.33 mM for RRM1-ZnF1$_S$ bound to RNA complex on AVIII600 Bruker NMR spectrometer and for RRM1-ZnF1$_S$-RRM2 free and bound to RNA at 0.3 mM on AVIII800 Bruker NMR spectrometer. For wild-type RRM1-ZnF1 and C191G mutant, RRM1-GGS-ZnF1, RRM1-ZnF1$_S$-L1, and RRM1-ZnF1$_S$-RNA complex, $^{15}N$ $R_1$ data were measured with 10 different relaxation delays and two duplicate delays, 21.6/21.6, 86.4, 162, 248.4, 345.6, 518.4, 669.6, 885.6/885.6, 1144.8, 1382.4 ms and $^{15}N$ $R_{1\rho}$ data were determined by using 10 different delay points with two duplicate delays, 5/5, 10, 15, 20, 40, 80, 100/100, 130, 160, 180 ms. For RRM1-ZnF1$_S$-RRM2 in free form, $^{15}N$ $R_1$ data were measured with 10 different relaxation delays and two duplicate delays,

21.6/21.6, 86.4, 162/162, 432, 540, 675, 810, 1080, 1350, 1620 ms and $^{15}N$ $R_{1\rho}$ data were determined by using 12 different delay points with two duplicate delays, 5/5, 10, 15, 20, 30, 50, 75, 80, 100/100, 115, 130, 160 ms. For RRM1-ZnF1$_S$-RRM2 - GGCU_12 RNA complex, $^{15}N$ $R_1$ data were measured with 11 different relaxation delays and one duplicate delay, 0, 80, 160, 240/240, 400, 560, 800, 960, 1200, 1440, 1600 ms, and $^{15}N$ $R_{1\rho}$ data were determined by using 11 different delay points with one duplicate delay, 5, 10, 15, 20/20, 30, 40, 50, 60, 80, 100, 120 ms. Duplicate time points were used for error estimation. The transverse relaxation rate $R_2$ for each residue was estimated by correction of the observed relaxation rate $R_{1\rho}$ with the offset Δν of the radio-frequency field to the resonance using the relation $R_{1\rho} = R_1 \cos^2\theta + R_2 \sin^2\theta$, where $\theta = \tan^{-1}(\nu_1/\Delta\nu)$. The correlation time ($\tau_c$) of the protein molecule was subsequently estimated using $R_2/R_1$[66]. The experiments were acquired as pseudo-3D experiments and converted to 2D data sets during processing in NMRPipe[59]. The relaxation rates and error determination were performed by using PINT[67]. Cross-peaks with low intensity or extensive overlaps were removed from the data analysis.

**Residual dipolar coupling data.** For all residual dipolar coupling (RDC) measurements, a 6% $C_{12}E_6$/hexanol stock was prepared in buffer supplemented with 10 % $D_2O$[68] and subsequently mixed in a 1:1 ratio with $^{15}N$-labeled 0.4 mM RRM1-ZnF1$_S$. The dipolar couplings were extracted from 2D in-phase–anti-phase (IPAP) HSQC experiments recorded under both isotropic and anisotropic conditions[69]. Long-term stability of the alignment, was assessed by comparing the $^2H$- quadrupolar splitting of $D_2O$ cosolvent before and after the experiment. NMR spectra were processed with NMRPipe[59] and the signal splittings were extracted from peak positions in CCPN Analysis. Only residues forming secondary structure or involved in $Zn^{2+}$ coordination (in case of ZnF1) were used for further analysis. PALES software[70] was used for the analysis of RDCs whereby the magnitude of alignment tensor ($D_a$) and rhombicity ($R$) were calculated using the principal components of traceless matrix ($A_{xx}$, $A_{yy}$, $A_{zz}$)-given by PALES and the absolute value of RDC. The Cornilescu $Q$ factor was used to determine the quality of the fit of experimental versus back-calculated RDCs[71].

**NMR titration experiments.** NMR RNA titrations were performed by adding 0.5, 1, 2-fold of GGCU_12 RNA into 0.1 mM $^{15}N$-labeled RRM2 or 0.5, 0.75, 1-fold of CU_9 into 0.1–0.2 mM $^{15}N$-labeled RRM1$_{F142A/F144A}$ and recording $^1H$-$^{15}N$ HSQC spectra in SEC buffer 2, supplemented with 10% $D_2O$. Due to low stability of free RRM1-ZnF1$_S$, RRM1-ZnF1$_S$-L1, and RRM1-ZnF1$_S$-RRM2 at 100 mM NaCl, GGCU_12 RNA was added in excess to $^{15}N$-labeled 0.2 mM protein in buffer containing 400 mM NaCl. The protein–RNA complex was then diluted to a 100 mM NaCl concentration, and subsequently concentrated using a 0.5 ml Amicon centrifugal filter concentrator with 3.5 kDa cut-off. To obtain a comparable spectrum of the free protein at 100 mM NaCl, the protein sample was diluted to ≈0.05 mM concentration. CSPs were calculated for backbone amide peaks of 2D $^1H$-$^{15}N$ HSQC correlation experiments using the equation, $\triangle\delta_{N,H}(ppm) = \sqrt{\Delta\delta_H^2 + (\alpha.\Delta\delta_N)^2}$ where $\alpha$ is a scaling factor with a value of 0.2.

**Paramagnetic relaxation enhancement experiments.** For PRE measurements, a natural solvent-exposed cysteine residue was mutated to serine (C230S). The free radical 3-(2-Iodoacetamido)-PROXYL spin label was attached to specifically engineered cysteine residues in RRM1-ZnF1$_S$-RRM2 (S155C in RRM1 domain and T249C, S288C, T307C in RRM2 domain). $^1H$-$^{15}N$ HSQC spectra were recorded for 0.15-0.25 mM $^{15}N$-labeled protein with or without GGCU_12 RNA and used to obtain the ratio of peak intensities ($I_{para}/I_{dia}$) of the paramagnetic and diamagnetic (after addition of 10-fold excess of ascorbic acid) and transferred into distances restraints as described[47,72].

To model the complex, a template structure with the coordinates of the RRM1-ZnF1 RNA complex, the RRM2 NMR structure (PDB ID: 2LKZ), and all missing residues of the protein and RNA was generated. Since a high-resolution structure of RRM2-RNA complex is not available, chemical shift perturbations in $^{15}$N-labeled RRM2 upon titration of CU_9 were used to generate ambiguous distances between the RNA residues 6–12 of GGCU_12 (CUUCUCU) and the protein residues which satisfy a threshold criterion ($\Delta\delta > 0.08$): Asp231, Thr232, Ile233, Arg236, Ile238, Ile262, Ile265, Lys268, Arg274, Phe276, Phe278, and Ala313. This led to a total of twelve ambiguous restraints of $4 \pm 1$ Å between the heavy atoms of each of these twelve protein residues to any of the RNA residues 6–12.

The template generated using this approach was randomized by random rotations of the phi/psi angles of the linker L1 followed by a short three-step simulated annealing energy minimization. To avoid clashes between the structured parts at the end of the randomization process, a random rotation was only accepted for a residue if the van der Waals (vdW) interaction energy of the molecule did not increase by more than 10% from the starting vdW energy. This led to 1750 randomized structures with proper geometries out of 2000 randomization processes started. This random pool of structures was minimized in a second step by a longer three-step Cartesian simulated annealing protocol including the PRE and chemical shift perturbation distance restraints.

## X-ray crystallography

RRM1-ZnF1 protein crystallized at a concentration of 10 mg/ml in a drop containing 0.1 M BICINE pH 9.0, 20 % PEG 6000 at 4 °C as very thin joint needles. After buffer optimization and using an additive screen, the thin needles were optimized to obtain thin plates in buffer containing 0.1 M BICINE pH 8.5, 12% PEG 6000, 0.1 M cesium chloride. Crystals were flash frozen in mother liquor supplemented with 20% ethylene glycol. Several datasets for the crystals were collected at ID23-1 and ID23-2 beamlines at ESRF, Grenoble. Datasets from best diffracting crystals were then processed with XDS software package[73,74] and the structure was solved by Auto-Rickshaw platform using MR-SAD[75,76]. The missing residues were built using Coot model building software[77] with multiple rounds of model building and refinement with Refmac software[78] from CCP4 suite[79]. The RRM1-ZnF1$_S$ –RNA complex (GGCU_10 RNA) was prepared by addition of 1.2-fold molar excess of RNA over protein and subsequent purification over a size exclusion column to remove excess RNA. The complex crystals were obtained in 0.05 M sodium cacodylate pH 6.5, 18 mM CaCl$_2$, 2.7 mM spermine, 9% 2-propanol at 4 °C as thin joint needles. Addition of 10% glycerol produced thicker separated needles, which were flash frozen in mother liquor containing 30% glycerol. The dataset was collected at ID30b beamline at ESRF Grenoble, France, and the crystals diffracted to a resolution of 3.5 Å. The crystals showed severe radiation damage, pseudo translation, and twinning. The dataset was integrated in space group P2$_1$ and was solved by molecular replacement with the individual structures of RRM1 and ZnF1 domains from the RRM1-ZnF1 structure as search models. Four molecules are found in the asymmetric unit. To mitigate the effect of pseudo translation on the $R_{free}$ values, the $R_{free}$ set was chosen in thin shells during refinement in Phenix, and the structures were refined with grouped B-factors, non-crystallographic symmetry restraints, and considering the h, -k, -l twin law. Due to low resolution of the data and the poor electron density because of the crystal pathologies, we could confidently model RNA in only one of the molecules of the asymmetric unit. All structures were visualized using Pymol v2.3.1 or Chimera v1.12.

## Static light scattering

All measurements were made with a Malvern Viscotek instrument (TDA 305) connected to an Äkta purifier equipped with an analytical size-exclusion column (Superdex 75 10/300 GL, GE Healthcare) equilibrated in buffer containing 20 mM MES pH 6.5, 100 mM NaCl, 1 mM DTT at 4 °C. A sample volume of 100 µl containing about 2–4 mg/ml of RRM1-ZnF1$_S$-RRM2 with or without RNA was injected for each run. A 1.2-fold molar excess of GGCU_12 RNA was added to the protein sample followed by an incubation period of 30 min on ice before sample injection. Elution profiles were collected for 30 min with a flow rate of 0.5 ml/min and data were collected using absorbance UV detection at 280 nm, right-angle light scattering (RALS) and refractive index (RI). The molecular weights of separated elution peaks were calculated using OmniSEC software (Malvern). As a calibration standard, 4 mg/ml bovine serum albumin was used before all experiments.

## Small-angle X-ray scattering

SAXS measurements with RRM1-ZnF1$_S$ were performed at 25 °C on the BioSAXS beamline BM29 with a 2D Pilatus detector at the European Synchrotron Radiation Facility (ESRF), Grenoble. Fifteen frames with 1 sec/ frame exposure time were recorded, using an X-ray wavelength of $\lambda = 0.9919$ Å, in flow mode at concentrations ranging from 1 mg/ml – 8 mg/ml. The dedicated beamline software BsxCuBE was used for data collection and processing. 1D scattering intensities of samples and buffers were expressed as a function of the modulus of the scattering vector $Q = (4\pi/\lambda)\sin\theta$ with $2\theta$ being the scattering angle and $\lambda$ the X-ray wavelength. Downstream processing after buffer subtraction was done with PRIMUS[80]. $R_g$ was determined using Guinier approximation and from p($r$) curve and crystal structure validation was done using CRYSOL[81]. Measurements for RRM1-ZnF1$_S$-RRM2 in apo at concentrations of 0.5 mg/ml and 1 mg/ml and for the RNA-complex at concentrations ranging from 0.6 to 4.8 mg/ml were performed at 5 °C using Rigaku BIOSAXS 1000. A purification of the protein–RNA complex using size-exclusion chromatography was performed to remove excess RNA before measurement. Primary data processing was done with Rigaku SAXSLab v 3.0.1r1 with eight frames with 900 sec/ frame exposure time recorded and data treatment was done with PRIMUS. All data were plotted using GraphPad Prism v6.

## Isothermal titration calorimetry

ITC experiments were performed either with MicroCal iTC200 or PEAQ-ITC calorimeters (Malvern). Prior to recording data, the protein samples were dialyzed overnight in buffer containing 20 mM MES pH 6.5, 100 mM NaCl, and 2 mM BME. The cell was filled completely with 10–30 µM protein and depending on the affinity, the syringe was filled with different concentrations of the respective ligand, ranging from 100 to 300 µM. A series of 26 injections of 1.5 µl titrant or 39 injections of 1 µl were made into the protein at 25 °C. The data were processed with either Origin (iTC200) or PEAQ-ITC Analysis software (PEAQ-ITC). The data were fit to a one-binding site model.

## Ex vivo splicing assays

Transfection assays were carried using 400,000 HeLa or HEK 293T cells seeded in 35 mm diameter tissue culture plates. 20 ng of the RG6-NUMB minigene[23] were co-transfected with 2 µg of plasmids expressing either T7 beta-galactosidase as control or T7-RBM5 (wild type or mutants) using Lipofectamine 2000. RNA was isolated 24 h after transfection and the pattern of alternative splicing was assessed by RT (with oligo-d(T)/random hexamer) and AMV Reverse Transcriptase (Promega) and PCR with GoTaq (Promega) using the primers-RG6 S: GGATTACAAGGATGACGATGACAAGGG, RG6 AS: GTCACCTT-CAGCTTCACGGTGTTGTG. Inclusion/skipping products were detected at 414/270 bp. Image quantification was performed using Fiji (Image J). The expression of the proteins was assessed by western blot using antibodies raised against the T7 epitope (Novagen, 1:1000 dilution) and GAPDH (Abcam, 1:1000 dilution). The GADPH signal was detected using mouse HRP secondary antibody (Cytiva, 1:10,000 dilution).

## Reporting summary

Further information on research design is available in the Nature Portfolio Reporting Summary linked to this article.

## Data availability

The data supporting the findings of this study are available from the corresponding authors upon reasonable request. NMR backbone chemical shifts for RRM1 and RRM1-ZnF1$_S$-RRM2 have been deposited to the BMRB under the accession codes 51057 and 51058, respectively. Coordinates and structure factors for the RRM1-ZnF1 apo and RRM1-ZnF1$_S$ in complex with GGCU_10 have been deposited in the PDB with accession codes 7PCV and 7PDV, respectively. SAXS data for RRM1-ZnF1 apo, RRM1-ZnF1$_S$-RRM2 apo and in complex with GGCU_12 have been deposited to the SASBDB with accession codes SASDM43, SASDM53 and SASDM63, respectively. All source data are provided with this paper in a Source Data file. The NMR structure of RBM5 RRM2 used for generation of RRM1-ZnF1$_S$-RRM2 + RNA structure model is available in the PDB with accession code 2LKZ. Source data are provided with this paper.

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

## Acknowledgements

We thank Martin Rübbelke, Lisa Warner, André Mourão, and members of the Sattler lab for discussions, Debashish Ray and Timothy Hughes for sharing RNACompete data. K.S. is grateful for support from IMPRS-LS graduate school. P.K.A.J acknowledges support from a Boehringer Ingelheim Fonds doctoral fellowship. S.M.L. is supported by an EU HORIZON 2020 research and innovation program under the Marie Skłodowska-Curie grant agreement No. 792692. This work was supported by the German Research Foundation (DFG) grants SFB1035 (project number 201302640), GRK1721 (project number 178567888), and SPP1935 (project number 273941853) to M.S.

## Author contributions

K.S. performed NMR experiments, crystallographic structure determination, SAXS, and biophysical characterization. P.K.A.J. performed crystallographic structure determination. S.M.L. performed NMR experiments. S.B. performed ex vivo splicing assays. S.B. and J.V. analyzed splicing assays. A.G. recorded and analyzed SLS experiments. R.S. recorded SAXS experiments. B.S. performed structure modeling. K.S. and M.S. designed the study and wrote the manuscript. All authors commented and approved the final version of the manuscript.

## Funding

## Competing interests

The authors declare no competing interests.
