## [Peer Review File · Nature Communications]

Structural basis for specific RNA recognition by the alternative splicing factor RBM5REVIEWER COMMENTS

Reviewer #1 (Remarks to the Author):

This is an interesting manuscript which describes the interactions of the RBM5 RRM1-ZnF1-RRM2 domains to cis-regulatory RNA elements. The authors show the structure of the RRM1-ZnF1 region in complex with RNA in a non-canonical fashion and prove that, While the RRM1-ZnF1 domains act as a single structural module, RRM2 is connected by a flexible linker and tumbles independently. The subject is timely. My only criticism is in the presentation which is a bit chaotic. The introduction contains too many details. A shorter but more essential introduction would be appreciated. The figures are too overloaded. The authors should try to simplify them and in any case to recall them sequentially. The discussion should also be shortened.

Reviewer #2 (Remarks to the Author):

In this manuscript, Soni and co-workers discuss the molecular features of RNA recognition by the cancer protein RBM5. RBM5 is alternative splicing factor whose dis-regulation has been reported to be important in a number of common cancers. The main RNA binding region of the protein comprises three RNA binding motifs, two RRM domains and one Zinc finger. The authors examine inter-domain interactions and RNA binding, using NMR, X-ray crystallography, SAXS and ITC. The data indicate that the different domains collaborate to recognize RNA, with the two adjacent domains recognizing different moieties of the same nucleotide and the downstream RRM1 being only loosely coupled to this unit. Next, Soni and others solved the structure of the first two RNA binding domains, RRM1 and ZnF1, alone and in complex with an RNA sequence extracted from a known RNA targets site. The structure reveal a novel binding mode for both the ZnF and RRM domains, and the authors validate the protein residues mediating the interactions using a number of mutations. In addition, they obtain long range distance information and chemical shift perturbation information to create a structural model for the entire three domain unit in complex with RNA. Finally, they test the relevance of their findings using ex vivo splicing assays.

The paper is well written and the figures are very informative. The logic of the study is clear and the results are interesting and non-obvious. This paper represents a valuable contribution to the field. The conclusions are overall well supported by the data and well discussed – see specific comments below. There are a number of points that need to be addressed prior to publication.

Scientific comments:

- Page 4: the authors discuss that Collins et al report that the RBM10 PAR-CLIP-identified sequence includes mainly C-rich sequences, and that the recognition of a GGA sequence comes from an in vitro analysis. Instead, in the paper Collins et al, extract the CUGUGGA sequence from a re-analysis of PAR-CLIP targets, test it in vitro and validate using minigene assays. This should be corrected.
- Page 11: The authors mention that in Figure 1c the CS of RRM2 and RRM1 are very similar isolation and in the three-domains (RRM1-ZnF-RRM2) construct. While I can see this is the case for RRM2 I see

substantial shifts for RRM1, which is not surprising as the authors report an interaction.

- Page 12: The authors mention that the arrangement of the tandem di-domain is not maintained in the 3-domain construct as shown in Fig1d. However, the NMR CSP data show that, once the unfolded residues C-terminal to the ZnF are added the spectra look very similar. I am not sure what is the rationale to say the domains interact differently at this point of the study.

- In the sequence used in the cited RBM10 study, a number of nucleotides 5' of the GG motif are shown to be relevant to the binding. In order to exclude that differences in RBM5 and RBM10 RNA binding are not sequence-dependent, the authors need to test the affinity RBM5 for an oligo where the GG motif is not at the 5' - e.g. the sequence used for RBM10 or an extended (at the 5') version of the oligo in this paper.

- The limitations of the model that includes RRM2 could be discussed more.

Presentational comments:

- I would have Figure S6 as a main Figure.

- The legend of Figure S7 should read 'red and black', not 'blue and grey'.

Reviewer #3 (Remarks to the Author):

The manuscript by Soni et al. describes the results of structural and biophysical experiments by constructs derived from the multi-domain alternative splicing factor RBM5. This is one of several proteins that contain RRM domains, among others, that have been implicated in regulation of alternative splicing. While the work only characterizes three of the at least seven domains identified by sequence of the full-length protein, the combined NMR-crystallography-SAXS-ITC and mutational characterization is thorough, reveals a new modality of zinc finger-RRM cooperativity, and should be of interest generally to scientists who study RNA structure, RNA recognition, and RNA biology broadly. Thus, I support publication. I have some suggestions to improve the presentation.

1. An important limitation of the work is that the crystal structure of the RNA complex is at modest resolution, and more importantly, derives from crystals that suffer from a number of pathologies (pseudotranslation, twinning) and which were damaged by X-rays during data collection (as indicated in methods). Indeed, the authors state that of the four protein copies in the asymmetric unit, (by the way, it's "asymmetric unit", not "asymmetrical" as in line -3 of page 9) only one provided electron density sufficient to build the RNA. While this information is present in the methods section, I think the authors need to be more forthright in this regard. Thus I think at a minimum, a figure panel in the main text (that would be Fig 3, in the current MS) needs to show the unbiased (i.e., before the RNA was built) electron density map (an Fo-Fc map is probably best) for the single copy in the au that allowed building of the RNA model.

2. Likewise, for the experts, the methods section should indicate if twinning was used during refinement, and if so, which twin law(s) were involved. Was anything done to mitigate the effect of the pseudotranslation on the scattering intensities?
3. Also to allow the reader to better judge the reliability of the models overall, the estimated mean coordinate precision (Luzzati or similar) should be included in Table 1.
4. In Table 1, the significance of digits needs to be considered. I doubt very much that, for instance, the resolution can be cited meaningfully to extend to "3.495" Å. Given the likely precision of the atomic coordinates (above), "3.5" is about right.
5. Given the pathologies mentioned, and the resolution, I am surprised that the mean B-factor for the complex structure is only 31.5 Å². Are non-crystallographic symmetry restraints being used in refinement? Are the B-factors being refined as groups? More information in the methods section regarding crystallographic refinement would be valuable to evaluate potential limitations of the crystal structure.
6. In the same vein, please indicate by what criteria the hydrogen bonds in Fig 3 are being designated as such. I imagine the coordinate precision is of the order 0.4 Å, so are H-bonds considered all the way to 3.8 Å? Or is the cutoff 3.4 Å, or something else? What about the angle of the H-bonding groups?
7. Experimental (MR-SAD, prior to density modification) for the non-RNA crystal structure should be included in supplement.
8. Methods, for concentration calculation based on UV, what extinction coefficients were used for protein and for RNA?
9. Methods, page 9, is "10% 1M cesium chloride" actually 100 mM?
10. Methods page 10, Indicate the concentrations used (for protein and complex) for the two reported SAXS experiments.
11. Since the authors went to the trouble of performing calorimetry, please indicate explicitly the enthalpy of association (and if they want, the entropy, but at least with the enthalpy, the interested reader can calculate that). Likewise, for the ITC fits, indicate the incompetent fraction (inactive molecule %) in the fit.
12. In the discussion, it would be worth reminding the reader that the splicing experiments are with full-length protein, while the biophysics is with at most three domains (at least that is what I understood), and that therefore the conclusions regarding biological output have to consider the possible impact of the other domains.

13. SAXS, for all complexes, also provide the Kratky plots.

Point-by-point response

Reviewer #1:

This is an interesting manuscript which describes the interactions of the RBM5 RRM1-ZnF1-RRM2 domains to cis-regulatory RNA elements. The authors show the structure of the RRM1-ZnF1 region in complex with RNA in a non-canonical fashion and prove that, While the RRM1-ZnF1 domains act as a single structural module, RRM2 is connected by a flexible linker and tumbles independently. The subject is timely. My only criticism is in the presentation which is a bit chaotic. The introduction contains too many details. A shorter but more essential introduction would be appreciated. The figures are too overloaded. The authors should try to simplify them and in any case to recall them sequentially. The discussion should also be shortened.

Thank you for the appreciation of our study. We have rechecked the manuscript and shortened the introduction and discussion to focus on the most important aspects. We have also rechecked the figures and tried to simplify.

Reviewer #2:

In this manuscript, Soni and co-workers discuss the molecular features of RNA recognition by the cancer protein RBM5. RBM5 is alternative splicing factor whose dis-regulation has been reported to be important in a number of common cancers. The main RNA binding region of the protein comprises three RNA binding motifs, two RRM domains and one Zinc finger. The authors examine inter-domain interactions and RNA binding, using NMR, X-ray crystallography, SAXS and ITC. The data indicate that the different domains collaborate to recognize RNA, with the two adjacent domains recognizing different moieties of the same nucleotide and the downstream RRM1 being only loosely coupled to this unit. Next, Soni and others solved the structure of the first two RNA binding domains, RRM1 and ZnF1, alone and in complex with an RNA sequence extracted from a known RNA targets site. The structure reveal a novel binding mode for both the ZnF and RRM domains, and the authors validate the protein residues mediating the interactions using a number of mutations. In addition, they obtain long range distance information and chemical shift perturbation information to create a structural model for the entire three domain unit in complex with RNA. Finally, they test the relevance of their findings using ex vivo splicing assays.

The paper is well written and the figures are very informative. The logic of the study is clear and the results are interesting and non-obvious. This paper represents a valuable contribution to the field. The conclusions are overall well supported by the data and well discussed – see specific comments below. There are a number of points that need to be addressed prior to publication.

We thank the reviewer for these encouraging comments.

Scientific comments:

- Page 4: the authors discuss that Collins et al report that the RBM10 PAR-CLIP-identified sequence includes mainly C-rich sequences, and that the recognition of a GGA sequence comes from an in vitro analysis. Instead, in the paper Collins et al, extract the CUGUGGA sequence from a re-analysis of PAR-CLIP targets, test it in vitro and validate using minigene assays. This should be corrected.

Thank you for pointing this out. We have changed this to:

“Similar to RBM5, the tandem RRM1-ZnF1 domains of RBM10 have been shown to recognize a CUGUGGA-motif (Collins et al), while CLIP-Seq (Bechara et al) and PAR-CLIP (Collins et al, Wang et al) identified C-rich sequences shown to be preferentially bound by RRM2 (Collins et al).”

- Page 11: The authors mention that in Figure 1c the CS of RRM2 and RRM1 are very similar isolation and in the three-domains (RRM1-ZnF-RRM2) construct. While I can see this is the case for RRM2 I see substantial shifts for RRM1, which is not surprising as the authors report an interaction.

The spectral differences observed for both RRM1 and RRM2 are minor: The larger differences seen for RRM2 affect the boundary regions of the shorter construct. RRM1 does show basal chemical shift differences of around 0.1 ppm, which we agree with the reviewer reflects its coupling with ZnF1 in the three domain construct. We have added a statement to further clarify this:

“It is therefore not surprising that a comparison of ^1H - ^{15}N HSQC spectra of RRM1-ZnF_{1s}-RRM2 with the single RRM1 shows minor chemical shift differences all across RRM1 (Fig. S1c).”

- Page 12: The authors mention that the arrangement of the tandem di-domain is not maintained in the 3-domain construct as shown in Fig1d. However, the NMR CSP data show that, once the unfolded residues C-terminal to the ZnF are added the spectra look very similar. I am not sure what is the rationale to say the domains interact differently at this point of the study.

We believe that the interaction between RRM1 and ZnF1 in the tandem domain construct might at least in part reflect a truncation artifact as the construct is too short at the C-terminal boundary. This is why the addition of additional residues in the RRM1-ZnF_{1s}-L1 construct shows similar spectrum to the three-domain construct (RRM1-ZnF_{1s}-RRM2).

- In the sequence used in the cited RBM10 study, a number of nucleotides 5' of the GG motif are shown to be relevant to the binding. In order to exclude that differences in RBM5 and RBM10 RNA binding are not sequence-dependent, the authors need to test the affinity RBM5 for an oligo where the GG motif is not at the 5' - e.g. the sequence used for RBM10 or an extended (at the 5') version of the oligo in this paper.

Thank you for this comment. In order to clarify this, we have performed an ITC titration experiment following the reviewer's suggestion. The RRM1-ZnF_{1s} construct was titrated with an extended version of the oligo GGCU-12 at the 5'-end (5' - **ACUUGGCUCUUCUCU-3'**, henceforth referred to as extended GGCU_12- eGGCU_12), with the natural sequence of the caspase2 pre-mRNA. As seen in the Figure 1 for the reviewers in this response, we do not observe an increase in affinity when comparing with the original GGCU_12 ($K_D = 641 \pm 125$ nM for eGGCU_12 and $K_D = 263 \pm 27$ nM for GGCU_12), but rather find a reduction of the N-value and a small increase in the dissociation constant. This likely indicates a second suboptimal binding event upon an increase of the RNA length.

In any case this titration confirms that the nucleotides at 5' of the GG are not contributing towards the affinity of the protein for the RNA, contrary to what was observed for RBM10 (Collins *et al.*). These additional data are now added to **Suppl. Fig. 5k** and discussed in the text on page 13.

Figure 1 for the reviewers: Two replicates of the Isothermal Calorimetry Titration experiment as described in the methods section, using RBM5 RRM1-ZnF1_s (26.1 μM) in the cell and eGGCU_12 (205.6 μM) in the syringe. The obtained binding affinity, stoichiometry and enthalpies are $K_D = 641 \pm 125$ nM, $N = 0.61 \pm 0.02$ and $\Delta H = -61.7 \pm 2.9$ kJ/mol, respectively

- The limitations of the model that includes RRM2 could be discussed more.

We have added the following statement in the discussion

“Future studies should focus on providing high-resolution details for the contributions of all three domains for the recognition of extended RNA ligands and the specific roles of the L1 linker and RRM2.”

Presentational comments:

- I would have Figure S6 as a main Figure.

Thank you for this suggestion. We have moved panel b of old **Suppl. Fig. 6** to main Figure 4, panel d, and also included not just one experiment but the quantification of the biological replicates in panel e. Old Suppl Figure 6a was left in the supplement as it is just the protein expression control.

- The legend of Figure S7 should read 'red and black', not 'blue and grey'.

Corrected, in the new **Suppl. Fig. 9** of the revised manuscript.

Reviewer #3:

The manuscript by Soni et al. describes the results of structural and biophysical experiments by constructs derived from the multi-domain alternative splicing factor RBM5. This is one of several proteins that contain RRM domains, among others, that have been implicated in regulation of

alternative splicing. While the work only characterizes three of the at least seven domains identified by sequence of the full-length protein, the combined NMR-crystallography-SAXS-ITC and mutational characterization is thorough, reveals a new modality of zinc finger-RRM cooperativity, and should be of interest generally to scientists who study RNA structure, RNA recognition, and RNA biology broadly. Thus, I support publication. I have some suggestions to improve the presentation.

We thank the reviewer for the supportive comments and suggestions.

1. An important limitation of the work is that the crystal structure of the RNA complex is at modest resolution, and more importantly, derives from crystals that suffer from a number of pathologies (pseudotranslation, twinning) and which were damaged by X-rays during data collection (as indicated in methods). Indeed, the authors state that of the four protein copies in the asymmetric unit, (by the way, it's "asymmetric unit", not "asymetric" as in line -3 of page 9) only one provided electron density sufficient to build the RNA. While this information is present in the methods section, I think the authors need to be more forthright in this regard. Thus I think at a minimum, a figure panel in the main text (that would be Fig 3, in the current MS) needs to show the unbiased (i.e., before the RNA was built) electron density map (an F_o-F_c map is probably best) for the single copy in the au that allowed building of the RNA model.

Thank you for these comments. We fully agree and now provide the F_o-F_c map before the modeling of the RNA into the electron density as well as the $2F_o-F_c$ omit map for the RNA in **Suppl. Fig. 6**. The density of the RNA in the map before the modelling is indeed weak due to the pathologies in the crystal, and thus the F_o-F_c map is contoured at 1σ .

The typo regarding the asymmetric unit has been corrected.

2. Likewise, for the experts, the methods section should indicate if twinning was used during refinement, and if so, which twin law(s) were involved. Was anything done to mitigate the effect of the pseudotranslation on the scattering intensities?

We have added this information to the methods section. For the refinement, we used the h, -k, -l twin law. The R_{free} dataset was chosen in thin shells in Phenix during refinement to mitigate the effect of the pseudotranslation on the scattering intensities on the R_{free} set (although this did not improve too much). In addition, alternative possible space groups were tested but the current space group was found to give the best results.

3. Also to allow the reader to better judge the reliability of the models overall, the estimated mean coordinate precision (Luzzati or similar) should be included in Table 1.

We have included this information in Table 1 now. However, we are not sure if this is meaningful as the Luzzati criteria is obtained from the plot of R_{free} vs. resolution, and, given that R_{free} is elevated due to pseudotranslation, the calculated precision may not be very meaningful in this case.

4. In Table 1, the significance of digits needs to be considered. I doubt very much that, for instance, the resolution can be cited meaningfully to extend to "3.495" Å. Given the likely precision of the atomic coordinates (above), "3.5" is about right.

Thank you for pointing this out. We agree with the reviewer regarding the significance of the digits and have rounded them to the nearest decimal place in **Table 1**.

5. Given the pathologies mentioned, and the resolution, I am surprised that the mean B-factor for the complex structure is only 31.5 \AA^2 . Are non-crystallographic symmetry restraints being used in refinement? Are the B-factors being refined as groups? More information in the methods section regarding crystallographic refinement would be valuable to evaluate potential limitations of the crystal structure.

Indeed, due to the low resolution of the structure in addition to severe pathologies, which lead to further deterioration of the map quality, the B-factors were refined as groups and NCS restraints were used during refinement. We have added text explaining this to the methods section.

6. In the same vein, please indicate by what criteria the hydrogen bonds in Fig 3 are being designated as such. I imagine the coordinate precision is of the order 0.4 \AA , so are H-bonds considered all the way to 3.8 \AA ? Or is the cutoff 3.4 \AA , or something else? What about the angle of the H-bonding groups?

The hydrogen bonds were considered up to 4 \AA . This corresponds to the default criterium in Pymol of 3.6 \AA further relaxed by an additional 0.4 \AA reflecting the coordinate precision. To clarify this, we have labelled the distances between the atoms in the figure. For the hydrogen bond angles, the default value in Pymol (63°) was employed.

7. Experimental (MR-SAD, prior to density modification) for the non-RNA crystal structure should be included in supplement.

We assume the reviewer refers to the experimental map. We don't think that this map provides any information to the reader and so have not included in the manuscript. The experimental maps before and after density modification are shown below:

Before density modification,
contoured at 0.383 e/\AA^3

After density modification,
contoured at 0.19 e/\AA^3

8. Methods, for concentration calculation based on UV, what extinction coefficients were used for protein and for RNA?

We have added the molar extinction coefficients for both protein and RNA constructs in the Methods section.

9. Methods, page 9, is "10% 1M cesium chloride" actually 100 mM?

This is correct, it is now changed to 0.1 M cesium chloride.

10. *Methods page 10, Indicate the concentrations used (for protein and complex) for the two reported SAXS experiments.*

We have added the values in the methods section.

11. *Since the authors went to the trouble of performing calorimetry, please indicate explicitly the enthalpy of association (and if they want, the entropy, but at least with the enthalpy, the interested reader can calculate that). Likewise, for the ITC fits, indicate the incompetent fraction (inactive molecule %) in the fit.*

Thank you for this suggestion. The binding enthalpy and entropy are now provided in **Table 3**. For the ITC fits, assuming a 1:1 binding stoichiometry and the concentration of the RNA ligand in the syringe to be accurate, the N-value directly reports on the active concentration of the protein in the cell. The N values are provided in the Table 3 for all individual measurements. It has also been marked in the table whenever the N-value was fixed due to a low C-value titration.

12. *In the discussion, it would be worth reminding the reader that the splicing experiments are with full-length protein, while the biophysics is with at most three domains (at least that is what I understood), and that therefore the conclusions regarding biological output have to consider the possible impact of the other domains.*

As suggested by the reviewer we added the following statement in the discussion:

“We show that RRM1 and ZnF1 residues important for RNA binding also impact the activity of RBM5 in the NUMB exon 9 reporter assays. Note, that the in vitro studies focus only the RNA binding domains, while the splicing assays are performed with full-length protein, where additional domains present may contribute to splicing activity via protein-protein interactions.”

13. *SAXS, for all complexes, also provide the Kratky plots.*

We have included the Kratky plots in the **Suppl. Fig. 2 and 8**.

REVIEWERS' COMMENTS

Reviewer #2 (Remarks to the Author):

I look forward to see this study published.

Reviewer #3 (Remarks to the Author):

I think the revised ms is suitable for publication as is.